# Do ecological characteristics drive the prevalence of *Panulirus argus* virus 1 (PaV1) in juvenile Caribbean spiny lobsters in a tropical reef lagoon?

**Charlotte E. Davies¤, Patricia Briones-Fourzán ◉\*, Cecilia Barradas-Ortiz, Fernando Negrete-Soto, Gema Moo-Cocom, Enrique Lozano-Álvarez**

Universidad Nacional Autónoma de México, Instituto de Ciencias del Mar y Limnología, Unidad Académica de Sistemas Arrecifales, Puerto Morelos, Quintana Roo, Mexico

¤ Current address: Department of Biosciences, College of Science, Swansea University, Singleton Park, Swansea, Wales, United Kingdom.
\* briones@cmarl.unam.mx

## Abstract

PaV1 is a pathogenic virus found only to infect Caribbean spiny lobsters *Panulirus argus*, a major fishing resource. *P. argus* is a benthic mesopredator and has a complex life history, with several ontogenetic habitat changes. Habitat characteristics and species diversity of surrounding communities may have implications for disease dynamics. This is of more concern for juvenile lobsters, which are more susceptible to PaV1 and are far less mobile than adult lobsters. We targeted a population of juvenile *P. argus* in a reef lagoon in Mexico, where PaV1 was first observed in 2001. Prevalence has been since irregularly assessed, but in 2016 we began a more systematic assessment, with two sampling periods per year (June and November) in three different zones of the reef lagoon. To examine the relationship between PaV1 prevalence and potential ecological determinants, we assessed habitat complexity, cover of different substrates, and invertebrate community composition in all zones during the first four sampling periods (June and November 2016 and 2017). Habitat complexity and percent cover of some substrates varied with zone and sampling period. This was the case for seagrass and macroalgae, which nevertheless were the dominant substrates. The invertebrate community composition varied with sampling period, but not with zone. Probability of infection decreased with increasing lobster size, consistent with previous studies, but was not affected by zone (i.e., variations in ecological characteristics did not appear to be sufficiently large so as to influence prevalence of PaV1). This result possibly reflects the dominance of marine vegetation and suggests that lobsters can be sampled throughout the reef lagoon to assess PaV1 prevalence. Prevalence was higher in only one of seven sampling periods (November 2017), suggesting that the pathogen has leveled off to an enzootic level.

**Data Availability Statement:** All relevant data are within the manuscript and its Supporting Information files.

**Funding:** This project was funded by Universidad Nacional Autónoma de México-Dirección General de Asuntos del Personal Académico-Programa de Apoyo a Proyectos de Investigación e Innovación Tecnológica (UNAM-DGAPA-PAPIIT, http://dgapa. unam.mx/), Project IN-206117 granted to P.B.F., and a postdoctoral scholarship 2016-2018 also granted by DGAPA to C.E.D. The funders had no role in study design, data collection and analysis, decision to publish, or preparation of the manuscript.

**Competing interests:** The authors have declared that no competing interests exist.

# Introduction

The Caribbean spiny lobster *Panulirus argus* (Latreille, 1804) is an important fishing resource throughout the wider Caribbean region [1]. This species has a complex life history with several ontogenetic habitat changes. After an extended larval period that develops in oceanic waters for 5 to 7 months, the postlarvae of *P. argus* return to the coast and settle in shallow, vegetated habitats (seagrass meadows, macroalgal beds), where the small juveniles (6 to ~20 mm carapace length, CL, measured from the inter-orbital notch to the rear end carapace) dwell for approximately 2 to 4 months. Upon outgrowing the protection afforded by the vegetation, juvenile lobsters seek crevice-type shelters within or adjacent to the macroalgal beds or seagrass meadows. Later, the subadults (~50 to 80 mm CL) start migrating to coral reefs, which the adults (>80 mm CL) inhabit [2].

*P. argus* are omnivorous mesopredators and play an important ecological role in Caribbean coral reefs systems [3–5], but they are also susceptible to parasites and diseases [6]. For example, they are hosts to *Panulirus argus* virus 1 (PaV1), the first known naturally occurring virus of a lobster. PaV1 was first discovered in Florida (USA) in 2000 [7], and shortly thereafter in Puerto Morelos (México) in 2001 [8]. The main clinical/gross sign of infection is a 'milky' white hemolymph, immediately visible through the translucent membrane between the carapace and abdomen [7,9]. PaV1 only affects *P. argus* and is currently widespread throughout the Caribbean [10,11], linked to various factors. Principally, PaV1 prevalence has been correlated with lobster size, being found more in smaller, juvenile lobsters [12,13], and with habitat, being found more in highly-vegetated habitats [14–16], suggesting that vegetation may be acting as an environmental reservoir for the virus.

Although Caribbean spiny lobsters are gregarious, healthy lobsters tend to avoid heavily PaV1-infected conspecifics, which may help curb prevalence levels [17–20]. However, infections involve interactions not only between the pathogen and the host, but among networks of species [21]. Even in host-specific disease systems (such as PaV1/*P. argus*), species diversity of surrounding communities may affect disease dynamics in different ways [21–23]. For example, non-host species may reduce the probability of encounter between hosts, and if non-host species are prey or mutualists of hosts, they can reduce host stress, potentially increasing the efficacy of the host immune response [21]. Additionally, some non-host species may act as reservoirs of pathogens [24], although reservoirs of PaV1 and effects upon the ecosystem as a whole are unknown. As previously noted, some studies have reported a higher PaV1 prevalence in densely-vegetated areas compared to poorly-vegetated areas even after accounting for the significant effect of lobster size, suggesting that vegetation may be acting as an environmental reservoir for the disease [14,15]. This notion has been supported by a recent study [16] in which the probability of infection with PaV1 was found to be higher in lobsters inhabiting more vegetated habitats, but further proposing that either marine vegetation or fauna that live associated with vegetated habitats, or both, may be reservoirs of PaV1.

The present study aims to increase insight into the dynamics of the PaV1 disease in Caribbean reef lagoons, which are nursery habitats for juvenile *P. argus* [2]. We targeted the population of juvenile lobsters in the shallow Puerto Morelos reef lagoon (Mexico), where PaV1 has been present since 2001 [8]. Previously, prevalence of PaV1 in the reef lagoon had been assessed in irregular periods during 2005–2006 [13] and 2010–2014 [25]. Some assessments were based on lobster samples from specific sites within the lagoon and others on lobsters sampled throughout the reef lagoon. However, small-scale habitat characteristics (e.g. habitat complexity, types of substrate) and species diversity of local communities can play important roles in disease ecology [16,21,23,24]. This may be of more concern for juvenile lobsters than for adult lobsters because juveniles, especially those <50 mm CL, have far more limited movement ranges than adults [2,26,27] and are more susceptible to PaV1 [9,12,15]. Therefore, in 2016 we began a more systematic assessment,

with two samplings per year during contrasting seasons (June and November), in three zones of the reef lagoon. During the first four sampling periods (June and November 2016 and 2017), we also examined habitat characteristics, the composition of invertebrate communities (as a first step into an assessment of potential reservoirs of PaV1), and the size of lobsters in the different sampling zones to examine their potential relationship with PaV1 prevalence.

## Materials and methods

### Study area

The study was carried out in the Puerto Morelos Reef National Park, located at the northernmost section of the Mesoamerican Barrier Reef, on the Mexican Caribbean coast. This marine system (centered at 20˚52'N 86˚5'W) consists of an extended fringing reef located at a distance of ~0.5 to 2 km from the coast [28] (Fig 1). The reef reduces wave energy, allowing the presence of a shallow reef lagoon (~5 m in maximum depth) where seagrass meadows dominated by the turtlegrass *Thalassia testudinum* develop. The Puerto Morelos reef lagoon has been extensively studied since the early 1990s [29–36]. These studies have consistently divided the lagoon vegetation into three distinct zones: a narrow coastal fringe (50–100 m in width), a broad mid-lagoon zone, and a back-reef lagoon zone. In the mid-lagoon zone, which covers the greatest part of the lagoon, the sandy sediment tends to be deeper and the seagrass biomass and height are generally greater, but with substantial temporal and spatial variation [31,33]. In the back-reef zone, seagrass meadows have generally less biomass, shorter leaves, and a less dense canopy because the sediment layer is thinner and there is more hard substrate [33]. The present study took place in the mid-lagoon and back-reef lagoon zones.

The vegetation throughout the reef lagoon provides adequate settlement habitat for postlarvae and protection for small juvenile *P. argus* [37,38], but the abundance of large juveniles and sub-adult lobsters decreases abruptly because of the scarcity of crevice-type shelters in the lagoon [39]. At different times between 1998 and 2009, up to 80 experimental casitas (artificial shelters that mimic large crevices), scaled to harbor juvenile lobsters, were deployed on several sites throughout the lagoon to examine their long-term effects, first on density and biomass of juvenile lobsters, and later on PaV1 disease dynamics [13,25,27,40]. At the onset of the present study, 54 casitas remained operational. To examine whether variation in types of substrate, habitat complexity, and local invertebrate diversity were related with lobster size and prevalence of PaV1, we selected three zones in the reef lagoon where casitas were present but that differed in depth, density and height of seagrass (see [31,33]). Zone A, characterized by lower seagrass biomass with shorter leaves, was located near the back reef (~2.5 m in depth); zone B, characterized by higher seagrass biomass and canopy, was located in the mid-lagoon (3–3.5 m in depth); and zone C, with some characteristics similar to zone B, was located leeward of a reef channel, where the lagoon is broader and deeper (4.5–5 m in depth) (Fig 1). Distance between zones ranged between 600 m and 1 km. These distances exceed the typical movement ranges of juvenile *P. argus* ≤ 50 mm CL (<100 m), as previously assessed in this same reef lagoon [27].

### Ethics statement

A permit for sampling in the Puerto Morelos reef lagoon was issued by Comision Nacional de Acuacultura y Pesca, Mexico (PPF/DGOPA-259/14).

### Habitat characterization

To examine potential ecological differences among sampling zones and over time, percent cover of different types of substrate, habitat complexity, and invertebrate diversity were

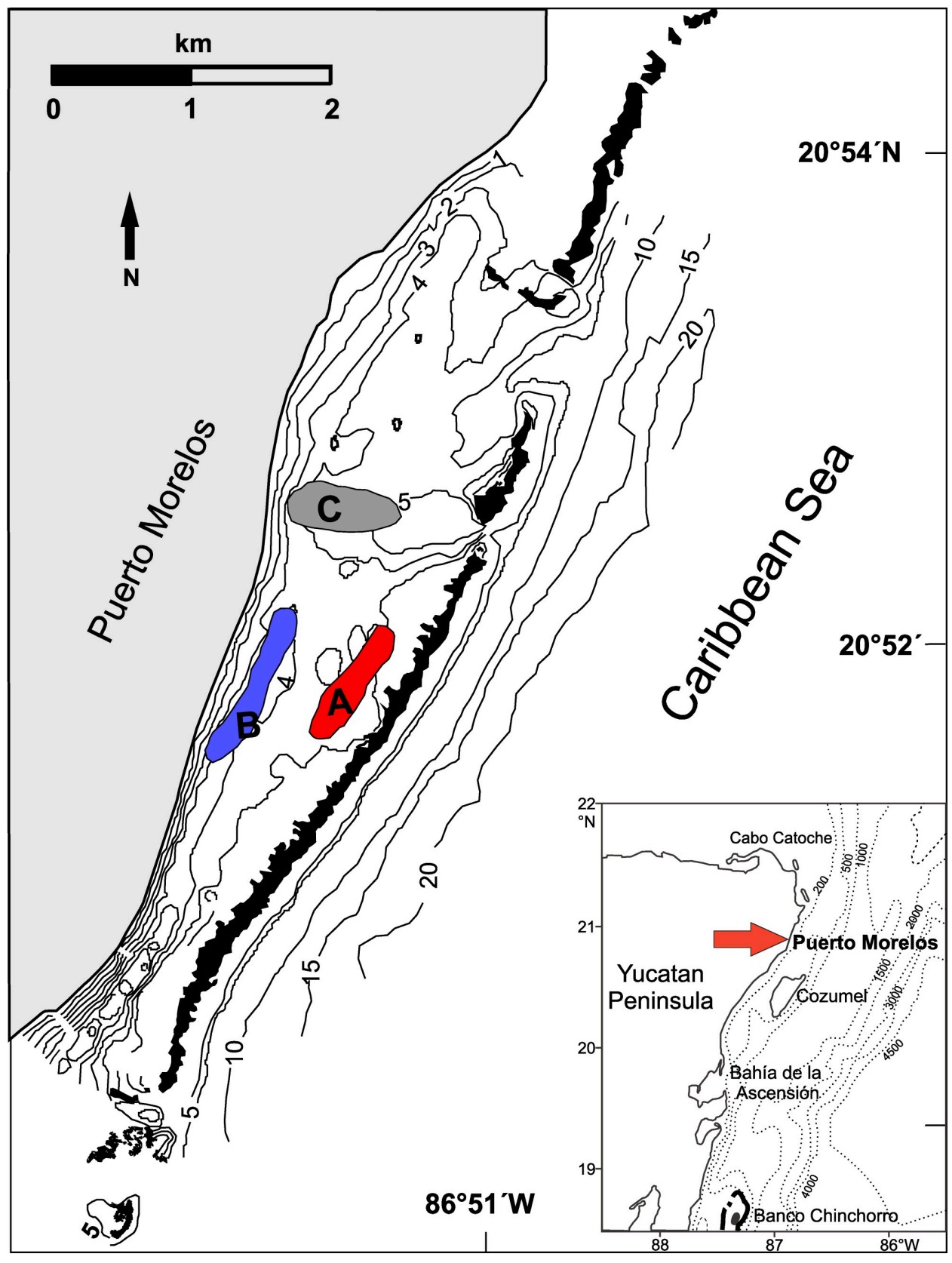

**Fig 1. Study area.** Location of the three sampling zones, zone A (red area), zone B (blue area), zone C (gray area), in the Puerto Morelos reef lagoon, Mexico. The black areas denote the reef crest. Isobaths are in meters. Inset shows the location of Puerto Morelos on the Mexican Caribbean coast. (Source: Servicio Académico de Monitoreo Meteorológico y Oceanográfico, Instituto de Ciencias del Mar y Limnología, Universidad Nacional Autónoma de México).

estimated in each zone [16] for two contrasting seasons (June and November) over two years (2016 and 2017), yielding four sampling periods. In each period, six transects, each 25 m in length, were laid parallel to the reef and the coast in each zone. Each transect was marked every 50 cm [16]. The percentage of cover of the following types of substrate was estimated using the point intercept method: seagrasses, macroalgae, sand, rubble, live hard corals, soft corals, hard bottom, and sponges [41]. This method consists of recording the type of substrate observed below each mark along a transect (= 50 estimates per transect). Since points are essentially dimensionless, the point intercept method is considered the least biased and most repeatable for determining cover [42].

Habitat complexity was estimated using a qualitative metric, the habitat assessment score (HAS) [16,43,44], which has the advantage that it can be applied in habitats from very complex (e.g. coral reefs) to very simple (e.g. sand) [43]. HAS provides an overall structural complexity value by visually evaluating six variables of the local topography (rugosity, variety of growth forms, height, refuge size categories, percentage of live cover, and percentage of hard substratum). Each variable is assigned a score between 1 and 5 (from smallest or lowest to largest or highest; see Table 1 in [43]), and the sum of the individual scores is the HAS. Therefore, a score of 6 would represent the least complex habitats and a score of 30 would represent the most complex habitats. Three quadrats, 2 m × 2 m each, were laid at the beginning, middle and end of each transect (N = 18 quadrats per zone). HAS was obtained within each quadrat by adding the scores of all components per quadrat.

### Invertebrate community composition

All conspicuous (> 1 cm) epibenthic macroinvertebrates (hereafter invertebrates) found within the same quadrats used to estimate HAS were identified to the lowest taxonomic level possible and quantified in situ [16,45]. Underwater identifications were conducted by two scientific observers thoroughly trained prior to sampling. Training was achieved by repeatedly studying an extensive guide of local invertebrate species created in our lab from photos and drawings obtained from many different sources, followed by direct identification in the field during preliminary dives, with the results being crosschecked between divers [16,45].

### Lobster sampling

In each zone, lobsters were sampled using scuba and free diving during the same four sampling periods as the habitat components and invertebrates, plus three additional periods (November 2018, and June and November 2019; logistic problems precluded sampling in June 2018). All lobsters encountered beneath casitas within each zone were collected with hand nets. Nets containing lobsters were fastened to the edge of the boat with lobsters remaining in the water to avoid exposure to air. Lobsters were sexed, measured (CL), and carefully examined for clinical signs of infection with PaV1 (milky hemolymph, visible through the translucent membrane between the carapace and abdomen) [9]. All lobsters with no clinical signs of PaV1 were returned to the capture site.

Two previous studies [15,46], one of them conducted in the same reef lagoon as the present study, established that, compared to endpoint polymerase chain reaction (PCR) assays [47], visual assessment of clinical signs of PaV1 had a specificity of 1 and a sensitivity of 0.5 (95%

CI: 0.4–0.6), meaning that a rough estimate of the true prevalence of PaV1 in this reef lagoon could be obtained by applying a 2x factor to the clinical prevalence estimated in a representative sample of lobsters. To corroborate those findings, ~200 μl hemolymph samples were taken from 402 lobsters sampled in the first two periods (June and November 2016). Hemolymph was taken from the base of the fifth pair of pereopods using a 30-gauge sterile needle and 1ml syringe, after disinfection of the puncture site with 70% ethanol. Hemolymph samples were fixed immediately in ice-cold 96% ethanol, transported to the laboratory and frozen at -20˚C.

## Hemolymph analysis of PaV1

**DNA extraction.** DNA was extracted from ~200 μl of hemolymph/ethanol mixture (~25 mg of hemolymph) with the Wizard genomic DNA purification kit (Promega) following a slightly modified manufacturer's protocol [16]. DNA eluted was used as the template for PCR. Hemolymph DNA extraction was optimized to ensure detection of PaV1 by using known, positive controls initially derived from *P. argus*. DNA extractions were verified by running 1 μl DNA mixed with 4 μl Promega Green GoTaqⓇ 5 x Flexi Buffer on a 1.5% TAE agarose gel.

**PCR conditions.** All PCRs were carried out using primers synthesized by Sigma and performed on a [3]Prime Personal Thermal Cycler (Techne, UK) before being visualized on a 1.5% TAE agarose gel. To test for the presence of PaV1 in lobsters, a PCR was performed using known, specific primers for PaV1 (45aF: TTC CAG CCC AGG TAC GTA TC; and 543aR: AAC AGA TTT TCC AGC AGC GT) that amplify a region of 499 bp [47]. All PCR reactions were carried out in a total volume of 10 μl containing 1μl extracted DNA (50-200ng/ μl), 0.33 mM of each primer 45aF and 543aR, 2.5 mM MgCl2 (Promega), Green GoTaqⓇ 5 x Flexi Buffer (Promega), 0.4 mM dNTP mixture (Promega), and 0.75 u GoTaqⓇ Flexi DNA Polymerase (Promega) [16].

## Statistical analyses

**Habitat characterization.** For each sampling period, data on the percent cover of each type of substrate were logit-transformed [48] and subjected to independent general linear models (GLM) with sampling zone (3 levels) and period (4 levels) as independent variables [49]. The transformed data were then subjected to separate principal component analysis (PCA) using the software PAST v.3.09 [50]. The data on structural complexity (HAS score) were also subjected to a GLM with zone and period as independent variables. Significant results of GLMs were followed by Tukey's HSD multiple comparisons test. For these analyses, the software Statistica v.10 (StatSoft, Inc., USA) was used.

**Invertebrate community composition.** For each sampling zone and period, the following ecological indices were estimated: species richness (S, number of species), Shannon-Wiener's diversity (H'), Pielou's evenness (J') and Simpson's dominance (D). S is an informative index as it constitutes the basis of biodiversity estimates, whereas H', J' and D are compound indices (i.e., indices that combine species richness and abundance) hence providing a greater ability to discriminate sites [51]. Each index was subjected to a GLM with sampling zone and period as independent variables.

Differences in invertebrate community composition between zones were analyzed by non-metric multidimensional scaling (nMDS) using the Bray-Curtis similarity measure on a square-root transformation of the abundance data [52]. This transformation retains the quantitative information while down-weighing the importance of the highly abundant species [53]. The significance of the observed differences among zones was further tested with a one-way analysis of similarity, which provides an R-value, typically between 0 and 1. Values close to 0 are indicative of little difference whereas values close to 1 are indicative of a large difference in

sample composition [53]. The software PRIMER 6 v6.1.9 (PRIMER-E Ltd) was used to carry out these analyses.

**Lobster size and PaV1 prevalence.** Data on lobster size were log-transformed and subjected to a GLM to examine the effects of sampling zone and period. Binomial logistic regression models with logit link functions [54] were used to determine whether specific predictor variables had a significant effect on the probability of finding lobsters clinically infected with PaV1. In the first model, the predictor variables were size (CL, covariate), zone, and sampling period. Based on the results of this model, the second logistic model examined the effects of size (covariate) and sampling period only. Clinical prevalence (the percentage of clinically infected lobsters) was estimated for each sampling period, and 95% confidence intervals were computed using Wilson's score method with continuity correction [55]. These analyses were run in the software Statistica v.10.

## Results

### Habitat characterization

Of the eight types of substrate considered, only five (seagrasses, macroalgae, sand, rubble, and sponges) yielded sufficient data for the GLM analyses. Of these substrates, the percent cover of seagrass, macroalgae and sponges varied significantly with zone and period, with no significant interaction; the percent cover of rubble varied with zone and period but with a significant interaction, whereas the percent cover of sand was not affected by zone, period, or their interaction (Table 1).

The relative cover of types of substrate changed over time. The first two components in the PCA jointly explained 74% of the variance in June 2016, 78.1% in November 2016, 61.7% in June 2017, and 84.1% in November 2017 (Fig 2). In all periods, either the first or the second component was strongly defined by the percent cover of sand, rubble, and/or sponges, as these substrates exhibited large positive or negative loadings (denoted by the length of the corresponding green lines in Fig 2). In contrast, seagrass and macroalgae did not have large loading values in any period (Fig 2). This is because, despite significant spatial and temporal variation (Table 1), seagrass was the most abundant substrate on all three zones in all periods (39–73% cover), generally followed by macroalgae (12–28% cover) (Fig 3).

Habitat complexity (HAS values) varied significantly with zone (F = 13.30; df = 2, 204; p < 0.001) and sampling period (F = 3.34; df = 3, 204; p = 0.02), but the interaction term was not significant (F = 1.11; df = 6, 204; p = 0.832). HAS values differed significantly among all three zones, being lower in zone A (12.0 ± 0.3, mean ± 95% CI), intermediate in zone C (12.7 ± 0.4), and higher in zone B (13.4 ± 0.4) (Tukey HSD test on factor zone). Canopy height and size of refuges contributed to this difference because their mean scores were higher in zone B than in zones A and C. The only period with a significantly different overall HAS value was November 2016 (13.2 ± 0.5) (Tukey HSD test on factor sampling period), driven mainly by higher scores in zones B and C during that particular period (S1 Fig). The other three periods had lower HAS values (June 2016: 12.7 ± 0.4; June 2017: 12.4 ± 0.4; November 2017: 12.5 ± 0.5).

### Invertebrate community composition

In total, 5847 individuals belonging to 96 different invertebrate taxa were observed, including cnidarians, polychaetes, decapods, stomatopods, echinoderms, bivalves, and gastropods (S1 Table). Two of the five ecological indices (J' and D) did not vary with either zone or sampling period, whereas the other three (S, N, and H') varied significantly with sampling period but not with zone, and the interaction term was not significant (Table 2). In all three cases, June 2016 was responsible for the significant difference, as this period had lower values of S, N and

**Table 1. Effects of sampling zone and period on percent cover of substrate types.**

| Substrate type | Effect | DF | MS | F | p |
|---|---|---|---|---|---|
| Seagrass | Intercept | 1 | 14.978 | 379.231 | <0.001 |
| | Zone | 2 | 0.208 | 5.265 | 0.008 |
| | Period | 3 | 0.481 | 12.185 | <0.001 |
| | Zone × Period | 6 | 0.033 | 0.847 | 0.541 |
| | Error | 60 | 0.039 | | |
| Macroalgae | Intercept | 1 | 177.336 | 1965.22 | <0.001 |
| | Zone | 2 | 0.881 | 9.761 | <0.001 |
| | Period | 3 | 0.279 | 3.093 | 0.034 |
| | Zone × Period | 6 | 0.108 | 1.194 | 0.322 |
| | Error | 60 | 0.090 | | |
| Sand | Intercept | 1 | 302.069 | 717.209 | <0.001 |
| | Zone | 2 | 0.351 | 0.832 | 0.440 |
| | Period | 3 | 0.987 | 2.338 | 0.083 |
| | Zone × Period | 6 | 0.378 | 0.898 | 0.503 |
| | Error | 60 | 0.421 | | |
| Rubble | Intercept | 1 | 800.141 | 2536.957 | <0.001 |
| | Zone | 2 | 5.148 | 16.321 | <0.001 |
| | Period | 3 | 2.677 | 8.488 | <0.001 |
| | Zone × Period | 6 | 0.915 | 2.901 | 0.015 |
| | Error | 60 | 0.315 | | |
| Sponges | Intercept | 1 | 824.315 | 2153.463 | <0.001 |
| | Zone | 2 | 3.940 | 10.293 | <0.001 |
| | Period | 3 | 4.140 | 10.816 | <0.001 |
| | Zone × Period | 6 | 0.564 | 1.473 | 0.203 |
| | Error | 60 | 0.383 | | |

Results of GLMs (α = 0.05) on logit-transformed data of percent cover of five types of substrate on three sampling zones (A, B, C) in the Puerto Morelos reef lagoon in four sampling periods (June and November 2016, June and November 2017) (N = 6 transects per zone per period).

H'. Nonetheless, the nMDS 2D ordination plots showed great overlap in the community composition of all three zones in every period (Fig 4). The stress values were relatively high (0.16–0.19) in three of the four periods, but 3D ordination plots (not shown) with stress values of 0.11–0.14 corroborated the great overlap among zones. This was further confirmed by analysis of similarity tests, which yielded R values of 0.067 for June 2016, 0.046 for November 2017, 0.144 for June 2017, and 0.115 for November 2017. These results indicate a substantial level of similarity in the invertebrate community composition across zones and periods. Overall, the ten most abundant invertebrate taxa included four gastropod species: *Tegula fasciata* (N = 1047), *Smaragdia viridis* (N = 387), *Cerithium litteratum* (N = 386), and *Modulus modulus* (N = 279); four decapod species: the hermit crabs *Pagurus brevidactylus* (N = 757), *Clibanarius tricolor* (N = 436), and *P. annulipes* (N = 277), and the crab *Mithraculus sculptus* (N = 163), and two ophiurid species: *Ophioderma appressa* (N = 283) and *Ophioderma* sp. (N = 247) (S1 Table).

## Lobster size

In total, 1503 lobsters were sampled throughout the study period. Size of lobsters ranged from 9.2 to 73.0 mm CL, with an overall mean (± SD) of 29.5 ± 10.5 mm CL (Fig 5). Mean size of lobsters by sampling zone and period fluctuated between 26.0 mm CL and 38.3 mm CL (S2 Fig). Mean size was significantly affected by sampling zone (F = 14.585, df = 2, 1482,

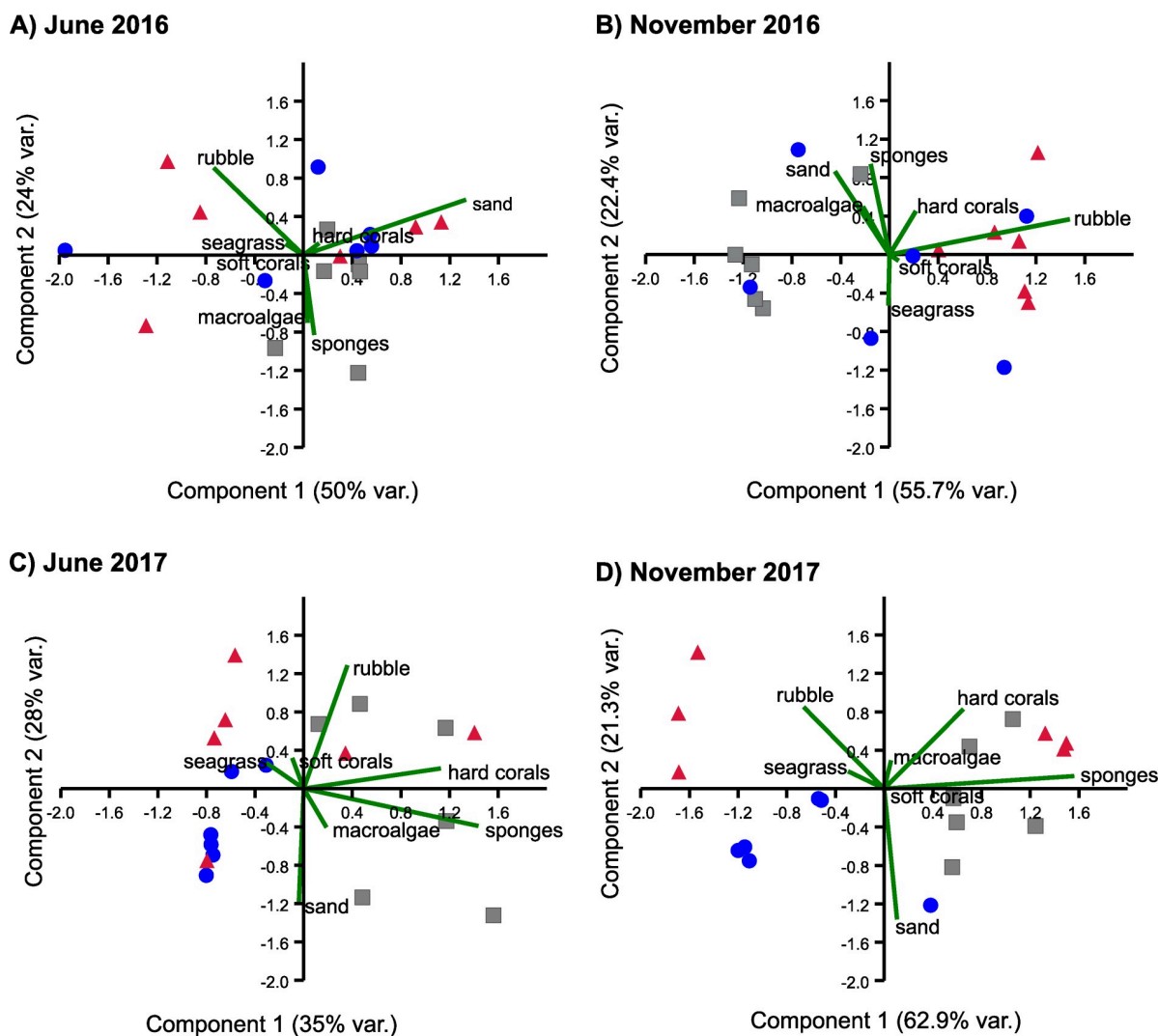

**Fig 2. Principal component analyses of percentage of cover of types of substrates.** Biplots on logit transformation of percentage of cover of seven types of substrate during four sampling periods (June and November 2016; June and November 2017) over three sampling zones within the Puerto Morelos reef lagoon, zone A (red triangles), zone B (blue dots), and zone C (grey squares). Each symbol represents a transect.

p < 0.001) and period (F = 16.488, df = 6, 1482, p < 0.001), with a significant interaction (F = 3.899, df = 12, 1482, p < 0.001). Mean size (± 95% CI) of lobsters was overall smaller in zone A (28.0 ± 0.96, N = 489), than in zones B (29.9 ± 0.84, N = 515), and C (30.2 ± 0.95, N = 499). Mean size of lobsters was smaller in November 2016, June 2017, and November 2017 than in the rest of the sampling periods

## Prevalence of PaV1

Of the total sample, 243 lobsters (16.2%) exhibited clinical signs of PaV1. These lobsters were relatively small, with a mean size of 27.2 ± 8.6 mm CL (size range: 10.4–60.3 mm CL). Prevalence values by individual zone and period varied from 5.4% to 27.3% (S3 Fig). However, in the first logistic regression model testing the effects of size, zone, and period on the probability of finding clinically PaV1-infected (i.e. diseased) lobsters, the effect of size was significant

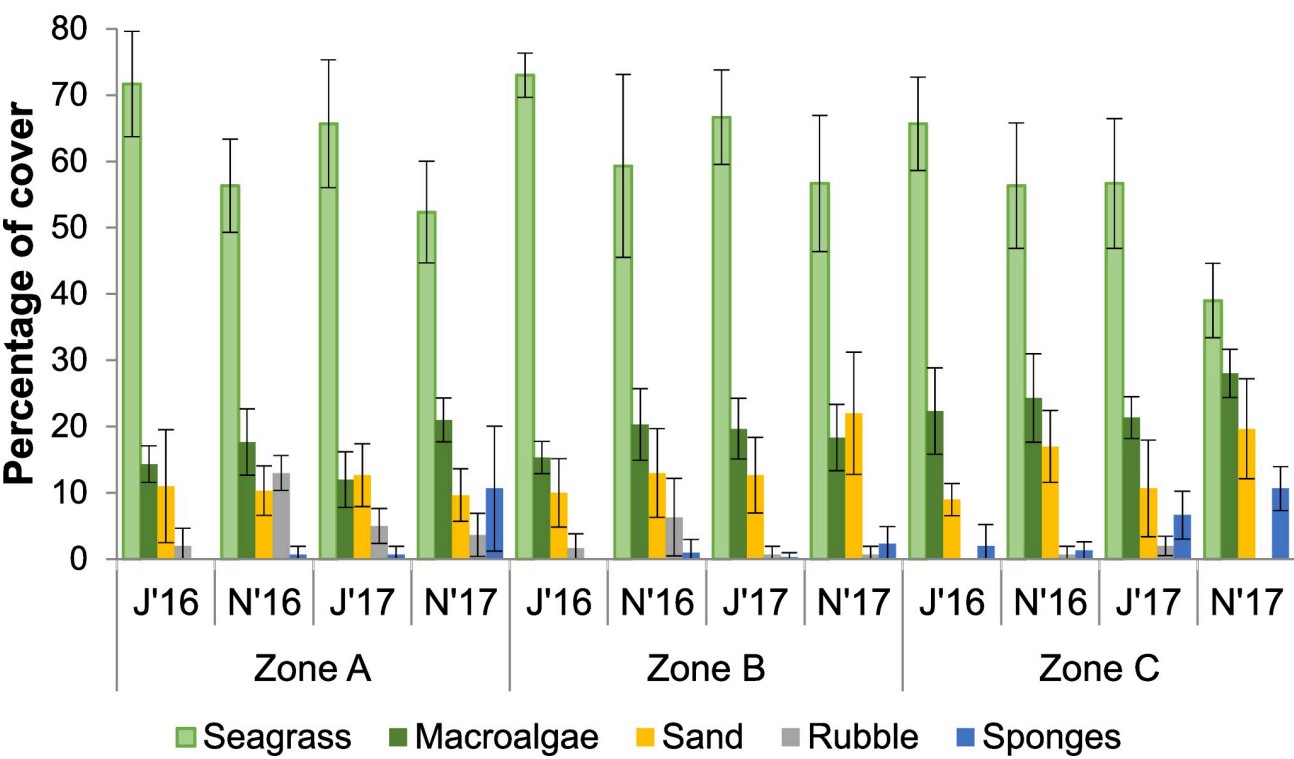

**Fig 3. Temporal and spatial variation in percentage of cover of benthic substrates.** Percentage of cover of the five most abundant substrates on the three sampling zones (A, B, and C) in the Puerto Morelos reef lagoon in four sampling periods: June 2016 (J'16), November 2016 (N'16), June 2017 (J'17) and November 2017 (N'17). Error bars denote 95% confidence intervals.

(Wald statistic, WS = 12.140, df = 1, p < 0.001), but the effects of zone (WS = 1.588, df = 2, p = 0.451) and period (WS = 9.922, df = 6, p = 0.128) were not significant. Parameter estimates of the model showed that the probability of finding clinically infected lobsters significantly decreased with increasing lobster size, and was slightly higher in November 2017 (Table 3). As we were particularly interested in examining the effect of time for monitoring purposes, we pooled the samples from the three zones by period and subjected the data to a second logistic regression model to examine only the effects of size and period on the probability of finding diseased lobsters throughout the reef lagoon. Parameter estimates from this model showed that the probability decreased with increasing lobster size and varied with sampling period, but was only significantly different (higher) in November 2017 relative to the other six periods (Table 4). Indeed, clinical prevalence of PaV1 was 22.5% in November 2017, compared to values between 13.4% and 18.6% in the other periods (Fig 6).

In the lobsters sampled for PCR assays in 2016, those testing positive for PaV1 amounted to 35.6% in June and 25.8% in November. Therefore, the proportion of lobsters with clinical signs of PaV1 relative to those testing positive for PaV1 by PCR was 0.43 in June 2017 and 0.57 in November 2017. These proportions are within the 95% confidence interval estimated for the sensitivity of clinical signs as compared to PCR assays [46], suggesting that the true prevalence of PaV1 across the entire study period may have varied between 26.8% and 45%.

## Discussion

Temporal variation in PaV1 prevalence in the Puerto Morelos reef lagoon was examined considering the potential influence of local habitat features and invertebrate community

**Table 2. Effects of sampling zone and period on ecological indices.**

| Ecological index | Effect | DF | MS | F | p |
|---|---|---|---|---|---|
| S | Intercept | 1 | 4617.37 | 929.935 | <0.001 |
| | Zone | 2 | 1.192 | 0.240 | 0.787 |
| | Period | 3 | 125.882 | 25.352 | <0.001 |
| | Zone × Period | 6 | 9.076 | 1.828 | 0.096 |
| | Error | 167 | 4.965 | | |
| N | Intercept | 1 | 19531.66 | 558.077 | <0.001 |
| | Zone | 2 | 58.230 | 1.664 | 0.193 |
| | Period | 3 | 1179.380 | 33.698 | <0.001 |
| | Zone × Period | 6 | 58.430 | 1.670 | 0.131 |
| | Error | 167 | 35.000 | | |
| H' | Intercept | 1 | 346.034 | 2191.343 | <0.001 |
| | Zone | 2 | 0.010 | 0.064 | 0.938 |
| | Period | 3 | 5.647 | 35.759 | <0.001 |
| | Zone × Period | 6 | 0.155 | 0.984 | 0.438 |
| | Error | 167 | 0.158 | | |
| J' | Intercept | 1 | 139.772 | 206066.400 | <0.001 |
| | Zone | 2 | 0.000 | 0.400 | 0.653 |
| | Period | 3 | 0.001 | 2.100 | 0.098 |
| | Zone × Period | 6 | 0.001 | 1.000 | 0.445 |
| | Error | 167 | 0.001 | | |
| D | Intercept | 1 | 116.249 | 21367.790 | <0.001 |
| | Zone | 2 | 0.002 | 0.380 | 0.685 |
| | Period | 3 | 0.002 | 0.390 | 0.764 |
| | Zone × Period | 6 | 0.002 | 0.330 | 0.919 |
| | Error | 167 | 0.005 | | |

Results of GLMs (α = 0.05) on data of five ecological measures of invertebrate diversity (S: species richness; N: abundance; H': Shannon-Wiener's diversity; J': Pielou's evenness; D: Simpson's dominance) in three sampling zones (A, B, C) in the Puerto Morelos reef lagoon during four sampling periods (June and November 2016, June and November 2017) (N = 18 quadrats per zone per period).

composition. Lobster size and the probability of clinical infection with PaV1 were inversely related, which has been previously well established (e.g. [9,10,12,14–16]). Although sampling period affected the probability of clinical infection, clinical prevalence was higher in only one of the seven sampling periods (November 2017). However, contrary to our expectations, zone had no effect on probability of infection. There were spatial and temporal variation in some of the ecological characteristics of the reef lagoon considered in the present study (e.g., habitat complexity, percent cover of different substrates), but such variations did not appear to be sufficiently large so as to influence prevalence of PaV1. This result probably reflects the dominance of marine vegetation (seagrass and macroalgae combined) in all three sampling zones and periods.

Because natural crevice-type shelters for lobsters are very scarce in the Puerto Morelos reef lagoon [39], the sampling zones included experimental sites where casitas were deployed years ago for other studies [25,27,39,40]. Casitas increase density of juvenile lobsters as well as their persistence in a site [27], and the distance between our sampling zones was greater than the average movement ranges of juvenile *P. argus* [2,26,27]. Yet, it cannot be dismissed that some mingling of lobsters could occur over time, potentially masking any effect of habitat characteristics on PaV1 prevalence. Therefore, future studies should use sampling sites that are further apart and, whenever possible, located over more heterogeneous habitats.

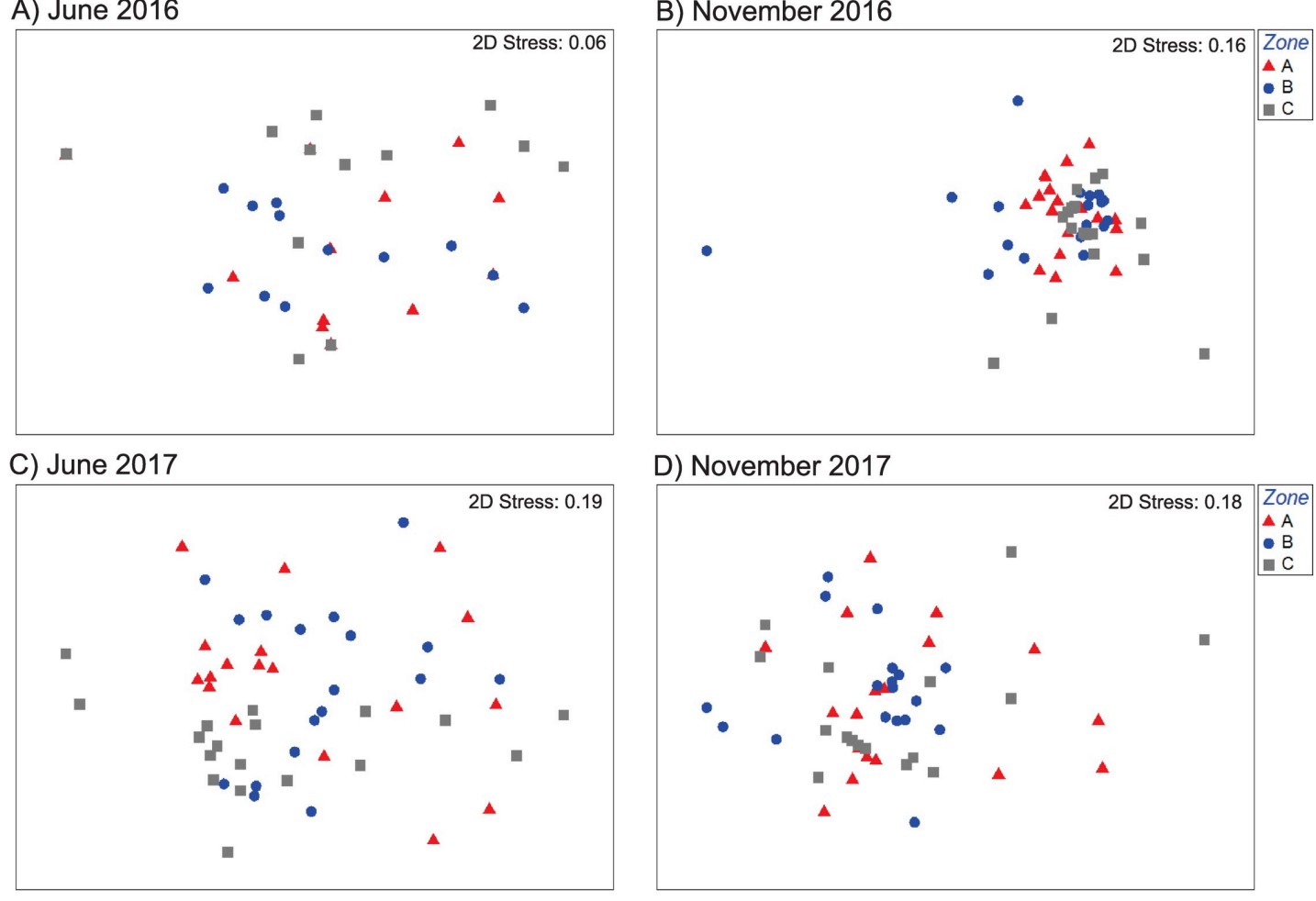

**Fig 4. nMDS ordinations of invertebrate communities.** nMDS ordination of invertebrate community structure in samples from zone A (red triangles), zone B (blue dots), and zone C (gray squares) of the Puerto Morelos reef lagoon in June 2016 (A), November 2016 (B), June 2017 (C) and November 2017 (D). Analyses were done using square-root transformation of species' abundances and Bray-Curtis similarity. Each symbol denotes a quadrat.

For example, habitat characteristics varied more substantially among sampling zones in Bahía de la Ascensión (México), a large bay about 150 km south of Puerto Morelos, where casitas are extensively used to fish for lobsters [9,14,15]. In that bay, probability of infection with PaV1 was higher in lobsters collected in a zone with more vegetation than in zones with less or no vegetation, even after controlling for the significant effect of lobster size, suggesting that marine vegetation could be an environmental reservoir for PaV1 [14]. This hypothesis was supported by a more recent study in the same bay, in which the probability of infection with PaV1 was highest in a reef lagoon zone dominated by seagrass, followed by a back-reef zone also dominated by seagrass but with less cover, and lowest in a zone almost devoid of vegetation, despite the lobsters in the latter zone having the smallest mean size [16]. Therefore, it would appear that the scale of habitat differences required to be associated with a change in disease prevalence requires a larger range of lobster sizes or a wider variation in habitats, such as those studied in Bahía de la Ascensión.

Certain crustacean viruses can remain infective in water for several days (e.g., invertebrate iridescent virus 6 [56]; yellow-head virus [57]), and waterborne transmission of PaV1 has been reported in juvenile *P. argus* held under laboratory conditions [10]. However, viruses, bacteria

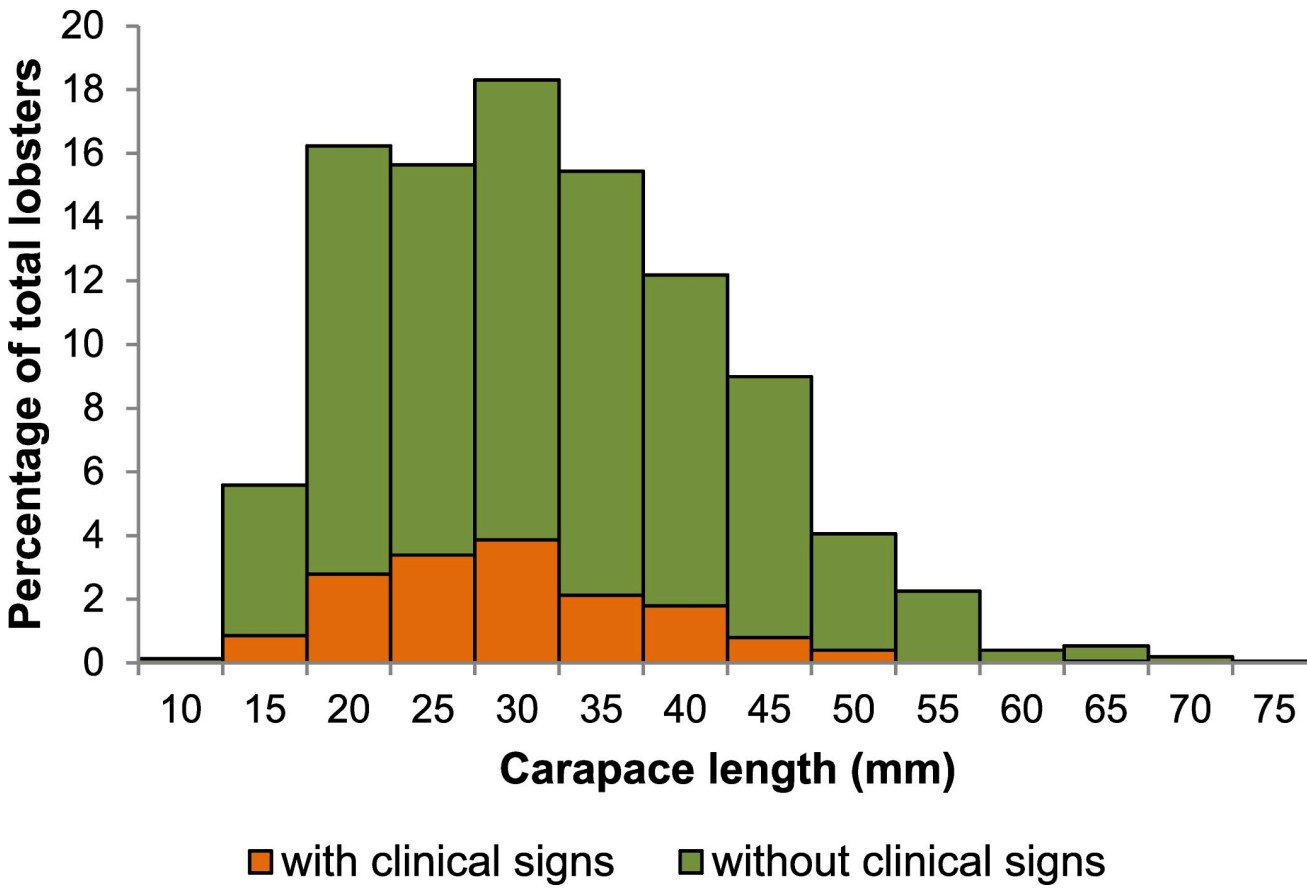

**Fig 5. Size distribution of lobsters.** Size distribution (carapace length, in mm) of the total sample (N = 1503) of spiny lobsters (*P. argus*) throughout the study. The orange column sections represent lobsters with clinical signs of PaV1 infection (N = 243) and the green column sections represent lobsters with no clinical signs of PaV1 infection (N = 1260). Numbers in X-axis denote the upper limit of each size class.

**Table 3. Results of logistic model 1.**

| Effect | Level of effect | Estimate | Standard error | Wald statistic | Lower 95% CL | Upper 95% CL | p |
|---|---|---|---|---|---|---|---|
| Intercept | | -0.926 | 0.227 | 16.655 | -1.371 | -0.481 | <0.001 |
| CL (mm) | | -0.026 | 0.008 | 12.140 | -0.041 | -0.011 | <0.001 |
| Period | Jun 16 | -0.015 | 0.231 | 0.004 | -0.467 | 0.437 | 0.948 |
| Period | Nov 16 | -0.117 | 0.162 | 0.521 | -0.436 | 0.201 | 0.470 |
| Period | Jun 17 | -0.242 | 0.174 | 1.921 | -0.583 | 0.100 | 0.166 |
| Period | Nov 17 | 0.362 | 0.184 | 3.895 | -0.002 | 0.722 | 0.048 |
| Period | Nov 18 | 0.266 | 0.157 | 2.871 | -0.042 | 0.574 | 0.090 |
| Period | Jun 19 | 0.050 | 0.226 | 0.048 | -0.393 | 0.492 | 0.826 |
| Zone | Zone B | -0.130 | 0.116 | 1.259 | -0.358 | 0.097 | 0.262 |
| Zone | Zone C | -0.016 | 0.114 | 0.020 | -0.208 | 0.241 | 0.885 |

Estimates for logistic regression analyses testing the effects of size (carapace length, covariate), sampling period (six levels: June and November 2016, June and November 2017, November 2018, June and November 2019; reference level: November 2019) and sampling zone (three levels: zones A, B, and C; reference level: zone A) on the probability of finding spiny lobsters *P. argus* clinically infected with PaV1 in the Puerto Morelos reef lagoon. CL: confidence limit.

**Table 4. Results of logistic model 2.**

| Effect | Level of effect | Estimate | Standard error | Wald statistic | Lower 95% CL | Upper 95% CL | p |
|---|---|---|---|---|---|---|---|
| Intercept | | -0.815 | 0.220 | 13.693 | -1.246 | -0.383 | <0.001 |
| CL (mm) | | -0.029 | 0.007 | 15.049 | -0.044 | -0.014 | <0.001 |
| Period | Jun 16 | -0.028 | 0.220 | 0.016 | -0.459 | 0.403 | 0.899 |
| Period | Nov 16 | -0.184 | 0.159 | 1.345 | -0.495 | 0.127 | 0.246 |
| Period | Jun 17 | -0.236 | 0.165 | 2.053 | -0.558 | 0.087 | 0.152 |
| Period | Nov 17 | 0.385 | 0.174 | 4.931 | 0.045 | 0.726 | 0.026 |
| Period | Nov 18 | 0.241 | 0.153 | 2.487 | -0.058 | 0.540 | 0.115 |
| Period | Jun 19 | -0.007 | 0.215 | 0.001 | -0.429 | 0.415 | 0.973 |

Estimates for logistic regression analyses testing the effects of size (carapace length, covariate) and sampling period (June and November 2016, June and November 2017, November 2018, June and November 2019; reference level: November 2019) on the probability of finding spiny lobsters *P. argus* clinically infected with PaV1 throughout the Puerto Morelos reef lagoon. CL: confidence limit.

and other particles can become trapped in seagrass meadows because the latter attenuate water flow velocity [58–60]. Bacteria and viruses can become adsorbed to plant surfaces [61], and although certain seagrasses produce natural bactericides [62] and seagrass meadows can

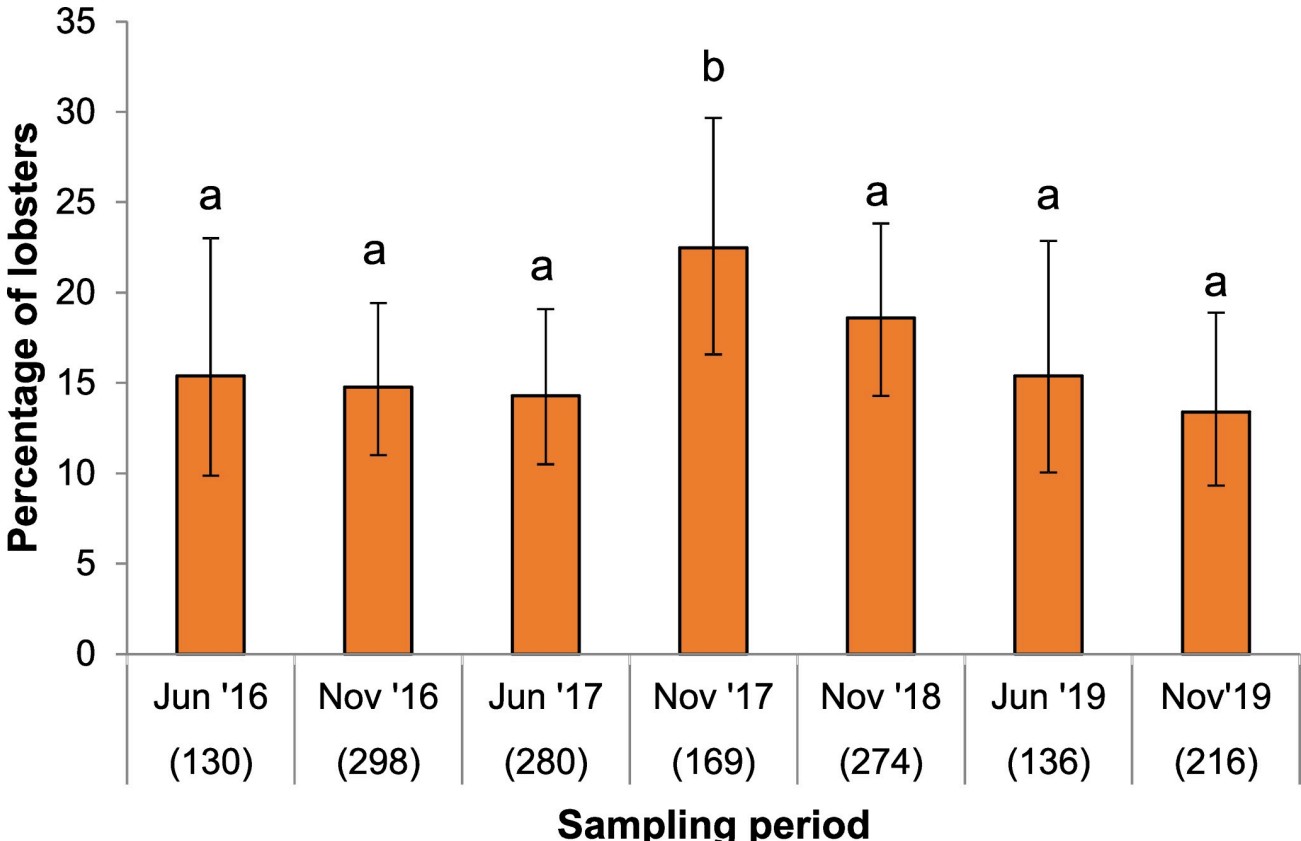

**Fig 6. Clinical prevalence of PaV1 over time.** Clinical prevalence of PaV1 (percentage of lobsters visibly infected, i.e., diseased) throughout the Puerto Morelos reef lagoon in seven sampling periods: June 2016 (Jun '16), November 2016 (Nov '16), June 2017 (Jun '17), November 2017 (Nov '17), November 2018 (Nov '18), June 2019 (Jun '19), and November 2019 (Nov'19). Numbers in parentheses are sample sizes. Error bars denote 95% confidence intervals. Different letters above bars denote significant differences.

reduce seawater pollution from human-originated bacteria [60], it has been suggested that the shading provided by the seagrass canopy may further protect virions from damaging ultra-violet radiation [14]. The presence of viable virions of PaV1 in seagrass meadows could be tested using environmental DNA techniques [63].

In the present study, the relatively high levels of clinical prevalence of PaV1 (13.4–22.5%) were clearly related to the small mean size of spiny lobsters in this reef lagoon, as juvenile *P. argus* are more susceptible to PaV1 than adults [7,13]. It was expected that the samples from all zones would comprise juveniles, since seagrass meadows constitute a nursery habitat for *P. argus* [2,37,38]. Although most of the lobsters that we sampled were found in experimental casitas, there is no evidence that the prevalence of PaV1 is any higher in areas where casitas are used [9,14,15,25]. This is probably because healthy lobsters avoid diseased conspecifics [17–20], and if forced to share a casita with a diseased lobster (e.g., due to the risk of predation), they tend to keep some distance from the diseased lobsters [40].

Other than a few crustacean pathogens such as the white spot syndrome virus, which has been confirmed to occur in many wild crustacean and non-crustacean species [24,64], little is known of the natural reservoirs and vectors of many crustacean pathogens. Such agents may play a critical role in the epizootiology and ecology of crustacean diseases but, to date, PaV1 has only been detected in *P. argus*. Butler et al. [10] inoculated hemolymph from PaV1-infected lobsters into multiple individuals of each of three crustacean species that live in sympatry with *P. argus*: the spotted spiny lobster *Panulirus guttatus*, the channel crab *Maguimithrax spinosissimus*, and the stone crab *Menippe mercenaria*. However, after several weeks, no histological evidence of PaV1 was found in any of these individuals. None of these crustacean species were observed in the sampling zones of the present study. If alternate hosts or vectors for PaV1 exist, they are more likely to be species that are syntopic with juveniles of *P. argus* [24,65], which exhibit higher levels of prevalence of PaV1 than adults.

In conjunction with a study conducted in Bahía de la Ascensión [16], one of the aims of the present study was to begin an assessment into potential reservoirs or vectors of PaV1 by identifying the invertebrate fauna living in the same habitats as *P. argus* in the Puerto Morelos reef lagoon as a first step. Although the composition of the invertebrate community varied significantly with sampling period, it did not vary with zone, as it did in Bahía de la Ascensión. Four of the 10 most abundant species in the Puerto Morelos reef lagoon were decapod crustaceans (*P. brevidactylus*, *C. tricolor*, *P. annulipes*, and *M. sculptus*). These species were also abundant in the back-reef and lagoon zones of Bahía de la Ascensión [16], making them good candidates for screening for PaV1 using molecular techniques, such as endpoint PCR [47] or qPCR [66].

Between 2000 and 2010, clinical prevalence of PaV1 in 12 sites of the Florida Keys fluctuated around an average of 5%, but varied both spatially and temporally, with some sites reaching >40% in a given year [12]. Also in the Florida Keys, mean yearly clinical prevalence of PaV1 fluctuated between 1 and 17% from 2005 to 2013 [19]. In the Puerto Morelos reef lagoon, the overall clinical prevalence of PaV1 increased from 2.7% in 2001, to 7.0% in 2005, to 10.9% in 2006 [13], and was found to fluctuate around a mean of 15% (95% CI: 10.8–18.8%) between 2010 and 2014 [25]. In the present study, clinical prevalence fluctuated around 16%, with only one estimate being significantly higher (22.5%, November 2017). Therefore, it is possible that in this location the pathogen has leveled off to an enzootic level [67,68]. However, as postlarvae of *P. argus* enter the reef lagoon throughout the year with great temporal variability [38] and some may become infected with PaV1 before settling [69,70], a certain amount of variation in the level of prevalence is to be expected.

Given that the specificity and sensitivity of the macroscopic determination of PaV1 estimated against endpoint PCR were 1.0 and 0.5, respectively [15,46], applying a 2x factor to clinical prevalence would provide a gross estimation of true prevalence. Therefore, true prevalence

of infection would fluctuate around a mean of ~32%. Although detection by PCR does not imply that all individuals testing positive would have active infections [47], it provides information about how widespread a virus is in a population [71].

Since the 1980s, the Puerto Morelos reef system has been gradually changing from a pristine system to a more eutrophic system, mainly due to continuous and sustained coastal development [31–35], which could be affecting the local biological communities. More recently, the tropical Atlantic and Caribbean Sea, including Puerto Morelos, are being impacted by massive influxes of the pelagic macroalgae *Sargassum* that, upon arriving to shallow near-shore seagrass communities, get stranded and die [72]. The decomposition of *Sargassum* masses produces a "brown tide" that severely depletes oxygen levels and reduces light penetration, killing the seagrass and changing the environmental conditions of the shallow habitats [72]. This is of concern for the biological communities of the reef lagoon, including the populations of juvenile *P. argus*, as the altered environmental conditions can cause mass mortalities of local fauna [73] and can also affect immunity either directly, by changing components of the immune responses, or indirectly, by inducing general stress responses [74]. According to recent studies, recurrent blooms of pelagic *Sargassum* in the tropical Atlantic and Caribbean Sea arrivals reflect a regime shift and may become the new norm [75]. Whether the changing environmental conditions associated with *Sargassum* strandings will alter the enzootic level of PaV1 in this population remains to be determined.

## Supporting information

**S1 Appendix. Data used in the article.**
(XLSX)

**S1 Fig. Temporal and spatial variation in habitat complexity.** Habitat assessment score in the three sampling zones (zone A: red columns; zone B: blue columns; zone C: gray columns) in four sampling periods: June 2016 (J'16), November 2016 (N'16), June 2017 (J'17) and November 2017 (N'17). Error bars denote 95% confidence intervals.
(PDF)

**S2 Fig. Lobster mean size by sampling zone and period.** Mean size (carapace length, mm) of lobsters sampled in three sampling zones (zone A: red columns; zone B: blue columns; zone C: gray columns) in the Puerto Morelos reef lagoon, in seven sampling periods: June 2016 (J'16), November 2016 (N'16), June 2017 (J'17), November 2017 (N'17), November 2018 (N'18), June 2019 (J'19), and November 2019 (N'19). Numbers in parentheses below dates are sample sizes. Error bars denote 95% confidence intervals. Different letters above bars denote significant differences.
(PDF)

**S3 Fig. Prevalence of PaV1 by sampling zone and period.** Percentage of lobsters showing clinical signs of PaV1 infection in three sampling zones (zone A: red columns; zone B: blue columns; zone C: gray columns) in the Puerto Morelos reef lagoon, in seven sampling periods: June 2016 (J'16), November 2016 (N'16), June 2017 (J'17), November 2017 (N'17), November 2018 (N'18), June 2019 (J'19), and November 2019 (N'19). Numbers in parentheses below dates are sample sizes. Error bars denote 95% confidence intervals.
(PDF)

**S1 Table. Invertebrate species list.** Invertebrate species (in alphabetical order within higher taxa) and number of individuals observed by sampling zone (zones A, B, and C) across four sampling periods (June and November of 2016 and 2017), Puerto Morelos reef lagoon (N = 18

quadrats, 2 m × 2 m, per zone per period).
(PDF)

## Acknowledgments

The authors would like to thank Roberto González-Gómez, Alí Espinosa-Magaña, Leslie Cid-González, and Raúl Tecalco-Rentería for assistance in field activities. We also thank Edgar Escalante-Mancera and Miguel A. Gómez-Reali from Servicio Académico de Monitoreo Meteorológico y Oceanográfico, Unidad Académica de Sistemas Arrecifales Puerto Morelos, Instituto de Ciencias del Mar y Limnología, UNAM, for generating the information to produce Fig 1.

## Author Contributions

**Conceptualization:** Charlotte E. Davies, Patricia Briones-Fourzán, Enrique Lozano-Álvarez.

**Data curation:** Charlotte E. Davies, Patricia Briones-Fourzán, Cecilia Barradas-Ortiz.

**Formal analysis:** Charlotte E. Davies, Patricia Briones-Fourzán, Gema Moo-Cocom, Enrique Lozano-Álvarez.

**Funding acquisition:** Patricia Briones-Fourzán, Enrique Lozano-Álvarez.

**Investigation:** Charlotte E. Davies, Patricia Briones-Fourzán, Cecilia Barradas-Ortiz, Fernando Negrete-Soto, Gema Moo-Cocom, Enrique Lozano-Álvarez.

**Methodology:** Charlotte E. Davies, Patricia Briones-Fourzán, Cecilia Barradas-Ortiz, Fernando Negrete-Soto, Gema Moo-Cocom, Enrique Lozano-Álvarez.

**Project administration:** Patricia Briones-Fourzán.

**Resources:** Patricia Briones-Fourzán, Enrique Lozano-Álvarez.

**Supervision:** Patricia Briones-Fourzán.

**Visualization:** Patricia Briones-Fourzán, Cecilia Barradas-Ortiz.

**Writing – original draft:** Patricia Briones-Fourzán.

**Writing – review & editing:** Charlotte E. Davies, Patricia Briones-Fourzán, Cecilia Barradas-Ortiz, Fernando Negrete-Soto, Gema Moo-Cocom, Enrique Lozano-Álvarez.

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
