## [Decision Letter · Decision Letter 0]

23 Oct 2019

PONE-D-19-25184

Monitoring the role of biodiversity and small-scale habitat change in disease prevalence among juvenile Caribbean spiny lobsters

PLOS ONE

Dear Dr. Briones-Fourzán,

Thank you for submitting your manuscript to PLOS ONE. After careful consideration, we feel that it has merit but does not fully meet PLOS ONE’s publication criteria as it currently stands. Therefore, we invite you to submit a revised version of the manuscript that addresses the points raised during the review process.

I found this to be an interesting and well written study. However, both reviewers have identified substantial issues with the manuscript that need to be rectified. The main issue with the study is that it seems the two sites were too homogenous to actually create any differences. Both reviewers have provided detailed comments to assist the authors in their revision and I have provided editorial comments. The authors should consider all the comments provided.

We would appreciate receiving your revised manuscript by Dec 07 2019 11:59PM. To enhance the reproducibility of your results, we recommend that if applicable you deposit your laboratory protocols in protocols.io, where a protocol can be assigned its own identifier (DOI) such that it can be cited independently in the future. For instructions see: http://journals.plos.org/plosone/s/submission-guidelines#loc-laboratory-protocols

We look forward to receiving your revised manuscript.

Kind regards,

Heather M. Patterson, Ph.D.

Academic Editor

PLOS ONE

**Journal Requirements:**

3. We note that  Figure(s) 1 in your submission contain [map/satellite] images which may be copyrighted. All PLOS content is published under the Creative Commons Attribution License (CC BY 4.0), which means that the manuscript, images, and Supporting Information files will be freely available online, and any third party is permitted to access, download, copy, distribute, and use these materials in any way, even commercially, with proper attribution. For these reasons, we cannot publish previously copyrighted maps or satellite images created using proprietary data, such as Google software (Google Maps, Street View, and Earth). For more information, see our copyright guidelines: http://journals.plos.org/plosone/s/licenses-and-copyright.

a) You may seek permission from the original copyright holder of Figure(s) [#] to publish the content specifically under the CC BY 4.0 license.  

**Comments to the Author**

1. Is the manuscript technically sound, and do the data support the conclusions?

Reviewer #1: No

Reviewer #2: Yes

2. Has the statistical analysis been performed appropriately and rigorously? 

Reviewer #1: Yes

Reviewer #2: Yes

3. Have the authors made all data underlying the findings in their manuscript fully available?

Reviewer #1: Yes

Reviewer #2: Yes

4. Is the manuscript presented in an intelligible fashion and written in standard English?

Reviewer #1: Yes

Reviewer #2: Yes

5. Review Comments to the Author

Reviewer #1: Overview: The manuscript provides an analysis of the habitat types, locations, and temporal components on the prevalence of PaV1 in spiny lobsters, Panulirus argus, from Puerto Morales, Mexico. It also attempts to associate possible differences in biodiversity between locations and prevalence. Although this was a large undertaking, it does present a few significant design flaws that detract from the findings. The locations selected for sampling were essentially too homogenous and so no differences were found among locations and few differences in temporal data. In effect, the locations A, B, and C are subsamples within the narrow confines of the reef-flat seagrass complex and have few differences. For example, seagrass was >73% cover in all transects and zones. That alone accounts for the lack of relationships found in the variables and their potential relationship with PaV1. Moreover, the postlarvae settle in seagrass and the early benthic juveniles and larger juveniles stay in this nursery habitat, leaving it as subadults or adults. Because the virus has a predilection for juveniles and juveniles prefer seagrass, the virus will be found in juveniles in seagrass. Thus relating the virus to seagrass habitat is a tautology, and thus not very interesting. This conclusion is addressed somewhat in the Discussion, but it would be better to be up front with this in the M&M, Results, and Discussion.

Comments to the Author

1. Is the ms technically sound and do the data support the conclusions?

The ms has a significant design flaw in that the locations selected for sampling were essentially subsamples. There was little heterogeneity in the habitat.

5. Review comments to the author

The title does not reflect the nature of the work that was done. The short title is better and I would place the habitat features first in the title. Monitoring wasn’t really the objective here and it’s overstated in the title. Another title could be “An investigation into the ecological determinants in the prevalence of PaV1 in juvenile lobsters from Puerto Morales, Mexico.”

The work on Cymatocarpus solearis does not contribute to the overall paper. I suggest dropping it because it’s superfluous. There weren’t enough infected lobsters to gain any additional understanding of this parasite in the lobster intermediate host.

Line 39: “infection” is misstated here. I would use the term “virus” or “pathogen” rather than infection. See also Line 489. The prevalence of the virus has fluctuated but is now established?

Biodiversity is mentioned in the title, but the effect of biodiversity is not mentioned in the abstract.

Line 61: garbled sentence. PaV1 only infects P. argus and is currently widespread…

Line 65 and in the Discussion: “vegetation may be acting as an environmental reservoir for the disease.” There are a few issues with this statement. First, the vegetation is likely not the reservoir, the reservoir is other infected lobsters shedding virus in this habitat. This could be tested by testing eDNA samples for the presence of the virus in the vegetated habitat as opposed to infected lobsters. This was not done. Second, disease is what happens in individual lobsters, the pathogen or agent is what is being sheltered or transmitted by a reservoir. Finally, lobsters with active, end-stage infections do not move much (morbidity) and thus wouldn’t be expected to move out of this habitat. The above points may be relevant to restructuring the Discussion.

Lines 89-92: again, the issue with the tautology.

Lines 93-103: these are presumably the objectives, but they are not well stated. They are too vague and loosely constructed to provide a logical flow to the ms. [Lines 472-473 give yet another objective. To tackle the notion of alternate hosts for the virus, one should “find and grind” many animals in the seagrass to see if any are overtly positive. One coudl also do more infection trials. These remain to be done in this system.]

Lines 68-79: delete this paragraph, superfluous.

Line 86: strike “relatively”. Ecosystem effects are unknown. Line 80: use a hyphen rather than a “/” when discussing host-parasite systems.

Line 143: why was 50 cm selected as the reference distance between points?

Lines 216-248: which variables are dependent variables and which are independent variables in the multitude of statistical analyses? Spell it out.

Lines 290, 293, 342, and elsewhere: Put emphasis on the finding, not on the statistic. Could rewrite as: Zone A had the lowest average HAS value and Zone B had the highest (ANOVA, Tukey’s HSD). “Results of GLMs showed that” this is an empty phrase. We’re interested in the biological finding.

Lines 361-368: delete this section on C. solearis. Data are too sparse. Pick up at line 369.

Line 404: the 95% confidence intervals: are these based on the quadrat data or the prevalence? If the latter then it’s a binomial (0/1) and the sd = square root (npq), thus the CIs would be smaller.

Discussion shows an important problem in the design on line 410: “seemingly different characteristics were selected”. A pilot study or earlier work in the region might have uncovered this issue. The zones weren’t different because the habitat was 73% seagrass in each zone; i.e., they're not very different.

Was temporal variation important? Was spatial variation important? Were there differences in this seemingly homogeneous habitat? These were investigated but they weren't presented well. A bigger question might be were their differences in lobster density within the zones and did this affect prevalence?

Lines 436-438: See above points regarding seagrass as an “environmental reservoir”. Also important would be host effects. I can think of many that could contribute to the host surviving longer in areas with more cover, easier access to food, etc. Host factors are important here and the predilection for the smaller animals may not be the only one.

Lines 510: not sure I agree with reference #28, stressors are known to increase pathogens that are host specialists as well as generalists. Vector-borne diseases are good examples. I think the issue is direct vs. indirect life cycles rather than host specificity. Suggest deleting or re-writing.

Table 2: the indices really should be labeled in the table heading.

Figure 2: not sure that this figure contributes much to the Results or Discussion, particularly given Figure 3. Suggest using one or the other, but not both. I’d go with Figure 3 and I’d probably use stacked bar graphs with the means data, but that may not be appropriate.

Figure 4: this might be best as supplemental data. There is very little variation in this data and I wonder if the statistical analysis had enough power to give any credence to the significant differences shown. This should be discussed.

Figure 5: I would use either Figure 4 or Figure 5 (I prefer Fig 5) to show that there is little separation in the habitats. Could do this for infected vs. uninfected animals and see if there are differences?

Figure 6: the clinical prevalence is lower than might be expected here, only hitting 4% at 30 mm CL. By the way, is a 73 mm CL lobster still a juvenile?

I would add an additional figure here. Shouldn’t there be a figure showing prevalence of PaV1 by site*time. (I was expecting something akin to Figure 7 but with prevalence data). I know this has been done in other studies of PaV1 from the region.

Figure 7 could be deleted with no loss to the main points.

Figure 8: see comments regarding the estimation of standard deviations from binomial data. I presume these are mean values from transects or zones, grouped and analyzed collectively. If that’s the case they are not binomial data, and the sample size isn’t reflective of the zone or transect number.

Minor style points:

Line 28: “and/or”. Just use one or the other, not both. “Or” usually works best.

Line 51: “structured crevice-type shelters”. This is jargon. It’s structured habitats or crevice-like shelters or dens.

Line 149: Re the habitat complexity estimates: these are probably more subjective than qualitative.

Line 162: give minimum size of animals identified in the biodiversity component. Was it >10 cm or >10 mm or some other value?

Line 199: American spelling was used throughout except for “Haemolymph”. Consistency.

Line 216: “percent data on the cover” is jargon. “data on the percent cover” is not. For statistical purposes, in logit-transformed data, I presume it’s presence/absence (0/1) data that are being analysed as in a logistic regression?

Line 233: why was square-root transformation used? I know why, but mention briefly to readers, i.e., to adjust variance to meet assumptions of normality.

Line 494-495: rewrite this sentence. There are well known means to adjust clinical prevalence levels using sensitivity and specificity.

Line 546: should spell out this journal reference.

Reviewer #2: Summary:

This manuscript describes an ambitious study aimed at determining whether there are habitat or community characteristics that can explain the prevalence of the virus PaV1 or the digenean trematode Cymatocarpus solearis among Caribbean spiny lobsters in a tropical reef lagoon offshore of Puerto Morelos, Mexico. The authors did not find any consistent associations between any of the characteristics they measured and the prevalence of these pathogens, other than one spike in PaV1 prevalence during one of the sampling periods.

General comments:

Overall the manuscript was well written and well organized. The abstract needs revision and the introduction needs some reorganization and revision (marked copy of these sections is attached), but the remainder of the manuscript text was clear and easily interpreted. The approach seemed appropriate, as were the statistical analyses and interpretation. The main problem with the manuscript is the lack of any consequential, significant findings. The only significant finding was a bump up in prevalence of PaV1 during one sampling period and this does not create any kind of pattern for interpretation (and the authors recognized this). This is always a difficult position to be in and I applaud the authors for writing the paper regardless because the lack of patterns or associations is still important to publish because, if for nothing else, it keeps other researchers for attempting similar studies that are apt to find the same results.

The sampling zones seem rather close to one another. I wonder if the proximity of the zones to one another doesn’t allow the lobsters to easily move between them and effectively ameliorate any effect of habitat or community characteristics on prevalence of these pathogens? I suggest that this be addressed in the discussion.

Specific comments:

PDF is attached with suggested edits to title, abstract, and introduction. Remainder of the manuscript was much better.

Note: ignore the bracketed comment on lines 68-79. At that point I thought the research was focused solely on PaV1 since C. solearis is not mentioned in the abstract at all (fix that)!

Line

172 Give a rationale for including the additional lobster sampling periods

It’s confusing the way the tables are nested in the manuscript with the captions following them, and the figure captions within the manuscript text. Are they not supposed to be after the literature cited?

Figures and table are nice and clearly constructed.

6. PLOS authors have the option to publish the peer review history of their article (what does this mean?). If published, this will include your full peer review and any attached files.

Reviewer #1: No

Reviewer #2: No

---

## [Author Response · Author response to Decision Letter 0]

3 Dec 2019

From: "PLOS ONE" <em@editorialmanager.com>

Subject: PLOS ONE Decision: Revision required [PONE-D-19-25184] - [EMID:78acbfad5d395c28] Date: Wed, October 23, 2019 12:26 am

To: Patricia Briones-Fourzán <briones@cmarl.unam.mx>

PONE-D-19-25184

Monitoring the role of biodiversity and small-scale habitat change in disease prevalence among juvenile

Caribbean spiny lobsters

PLOS ONE

Dear Dr. Briones-Fourzán,

Thank you for submitting your manuscript to PLOS ONE. After careful consideration, we feel that it has merit but does not fully meet PLOS ONE’s publication criteria as it currently stands. Therefore, we invite you to submit a revised version of the manuscript that addresses the points raised during the review process.

I found this to be an interesting and well written study. However, both reviewers have identified substantial issues with the manuscript that need to be rectified. The main issue with the study is that it seems the two sites were too homogenous to actually create any differences. Both reviewers have provided detailed comments to assist the authors in their revision and I have provided editorial comments. The authors should consider all the comments provided.

We would appreciate receiving your revised manuscript by Dec 07 2019 11:59PM. To enhance the reproducibility of your results, we recommend that if applicable you deposit your laboratory protocols in protocols.io, where a protocol can be assigned its own identifier (DOI) such that it can be cited independently in the future. For instructions see: http://journals.plos.org/plosone/s/submission-guidelines#loc- laboratory-protocols

A rebuttal letter that responds to each point raised by the academic editor and reviewer(s). This letter should be uploaded as separate file and labeled 'Response to Reviewers'.

A marked-up copy of your manuscript that highlights changes made to the original version. This file should be uploaded as separate file and labeled 'Revised Manuscript with Track Changes'.

An unmarked version of your revised paper without tracked changes. This file should be uploaded as separate file and labeled 'Manuscript'.

We look forward to receiving your revised manuscript. Kind regards,

Heather M. Patterson, Ph.D.

Academic Editor

PLOS ONE

Journal Requirements:

Our responses to comments are in green Calibri font. Please note that we numbered the reviewers’ comments to facilitate cross-referencing them. In our Responses, References which appear in the revised manuscript are followed by the corresponding number (e.g. [16]). References which do not appear in the References section of the revised manuscript appear at the end of this section of Responses.

R. PLoS One’s style requirements have been now carefully followed. Also, according to PACE, all figures meet PLOS requirements

R. Permit information has now been added, both in Methods and in Ethics Statement.

3. We note that Figure(s) 1 in your submission contain [map/satellite] images which may be copyrighted. All PLOS content is published under the Creative Commons Attribution License (CC BY 4.0), which means that the manuscript, images, and Supporting Information files will be freely available online, and any third party is permitted to access, download, copy, distribute, and use these materials in any way, even commercially, with proper attribution. For these reasons, we cannot publish previously copyrighted maps or satellite images created using proprietary data, such as Google software (Google Maps, Street View, and Earth). For more information, see our copyright guidelines: http://journals.plos.org/plosone/s/licenses-and-copyright.

a) You may seek permission from the original copyright holder of Figure(s) [#] to publish the content specifically under the CC BY 4.0 license.

We recommend that you contact the original copyright holder with the Content Permission Form

(http://journals.plos.org/plosone/s/file?id=7c09/content-permission-form.pdf) and the following text:

In the figure caption of the copyrighted figure, please include the following text: “Reprinted from [ref] under a

CC BY license, with permission from [name of publisher], original copyright [original copyright year].”

Maps at the CIA (public domain): https://www.cia.gov/library/publications/the-world-factbook/index.html and

https://www.cia.gov/library/publications/cia-maps-publications/index.html NASA Earth Observatory (public domain): http://earthobservatory.nasa.gov/ Landsat: http://landsat.visibleearth.nasa.gov/

USGS EROS (Earth Resources Observatory and Science (EROS) Center) (public domain): http://eros.usgs.gov/# Natural Earth (public domain): http://www.naturalearthdata.com/

R. Figure 1 is not reprinted from a copyrighted source; it was produced using original information generated in my own Institute by Servicio Académico de Monitoreo Meteorológico y Oceanográfico (SAMMO) (Academic Service of Meteorologic and Oceanographic Monitoring). This source has been added to figure caption. A content permission form signed by the officer in charge of SAMMO is being uploaded with the revised ms.

Comments to the Author

1. Is the manuscript technically sound, and do the data support the conclusions?

Reviewer #1: No

Reviewer #2: Yes

2. Has the statistical analysis been performed appropriately and rigorously? Reviewer #1: Yes

Reviewer #2: Yes

3. Have the authors made all data underlying the findings in their manuscript fully available?

Reviewer #1: Yes

Reviewer #2: Yes

4. Is the manuscript presented in an intelligible fashion and written in standard English?

Reviewer #1: Yes

Reviewer #2: Yes

5. Review Comments to the Author

Reviewer #1: Overview: The manuscript provides an analysis of the habitat types, locations, and temporal components on the prevalence of PaV1 in spiny lobsters, Panulirus argus, from Puerto Morales, Mexico. It also attempts to associate possible differences in biodiversity between locations and prevalence. Although this was a large undertaking, it does present a few significant design flaws that detract from the findings. The locations selected for sampling were essentially too homogenous and so no differences were found among locations and few differences in temporal data. In effect, the locations A, B, and C are subsamples within the narrow confines of the reef-flat seagrass complex and have few differences. For example, seagrass was >73% cover in all transects and zones. That alone accounts for the lack of relationships found in the variables and their potential relationship with PaV1. Moreover, the postlarvae settle in seagrass and the early benthic juveniles and larger juveniles stay in this nursery habitat, leaving it as subadults or adults. Because the virus has a predilection for juveniles and juveniles prefer seagrass, the virus will be found in juveniles in seagrass. Thus relating the virus to seagrass habitat is a tautology, and thus not very interesting. This conclusion is addressed somewhat in the Discussion, but it would be better to be up front with this in the M&M, Results, and Discussion.

R. We respectfully disagree that relating the virus to seagrass habitat is a tautology. Please see more extended response to Comment # 8 below.

Comments to the Author

1. Is the ms technically sound and do the data support the conclusions?

1. The ms has a significant design flaw in that the locations selected for sampling were essentially subsamples. There was little heterogeneity in the habitat.

R. We respectfully disagree. The locations selected for sampling were not essentially subsamples. Please see the more extended response to comment # 17 below.

5. Review comments to the author

2. The title does not reflect the nature of the work that was done. The short title is better and I would place the habitat features first in the title. Monitoring wasn’t really the objective here and it’s overstated in the title. Another title could be “An investigation into the ecological determinants in the prevalence of PaV1 in juvenile lobsters from Puerto Morales, Mexico.”

R. The title has been changed to: “Ecological determinants in the prevalence of PaV1 in juvenile Caribbean spiny lobsters in a tropical reef lagoon.”

3. The work on Cymatocarpus solearis does not contribute to the overall paper. I suggest dropping it because it’s superfluous. There weren’t enough infected lobsters to gain any additional understanding of this parasite in the lobster intermediate host.

R. Agree. The work on C. solearis has been removed from the manuscript.

4. Line 39: “infection” is misstated here. I would use the term “virus” or “pathogen” rather than infection. See also Line 489. The prevalence of the virus has fluctuated but is now established?

R. “infection” has been changed to “pathogen” and the sentence was rephrased to “suggesting that the pathogen has leveled off to an enzootic level”.

5. Biodiversity is mentioned in the title, but the effect of biodiversity is not mentioned in the abstract.

R. The title has been changed to: “Ecological determinants in the prevalence of PaV1 in juvenile Caribbean spiny lobsters in a tropical reef lagoon.”

6. Line 61: garbled sentence. PaV1 only infects P. argus and is currently widespread…

R. Sentence has been changed as suggested.

7. Line 65 and in the Discussion: “vegetation may be acting as an environmental reservoir for the disease.” There are a few issues with this statement. First, the vegetation is likely not the reservoir, the reservoir is other infected lobsters shedding virus in this habitat. This could be tested by testing eDNA samples for the presence of the virus in the vegetated habitat as opposed to infected lobsters. This was not done. Second, disease is what happens in individual lobsters, the pathogen or agent is what is being sheltered or transmitted by a reservoir. Finally, lobsters with active, end-stage infections do not move much (morbidity) and thus wouldn’t be expected to move out of this habitat. The above points may be relevant to restructuring the Discussion.

R. Marine vegetation can certainly be an environmental reservoir for bacteria and viruses (see Small and Pagenkopp 2013 [24]; Sweet et al. 2013). Viruses can adsorb to plant surfaces (Gerba 1984) [61] and shading can protect virions from UV radiation (e.g. Raymond et al. 2005). Indeed, eDNA could help test this hypothesis, but that was not an objective of this study.

8. Lines 89-92: again, the issue with the tautology.

R. We respectfully disagree that relating the virus to seagrass habitat is a tautology. It is true that the virus has a predilection for juveniles and that juveniles prefer seagrass; however, previous studies (Briones-Fourzán et al. 2012 [14]; Huchin-Mian et al. 2013 [15]; Davies et al. 2019 [16]) have shown that prevalence of PaV1 is significantly higher in seagrass habitats even after accounting for the significant effect of lobster size, i.e., that irrespective of size, lobsters have a higher probability of infection in seagrass habitats compared to other habitats. 

9. Lines 93-103: these are presumably the objectives, but they are not well stated. They are too vague and loosely constructed to provide a logical flow to the ms. [Lines 472-473 give yet another objective. To tackle the notion of alternate hosts for the virus, one should “find and grind” many animals in the seagrass to see if any are overtly positive. One coudl also do more infection trials. These remain to be done in this system.]

R. We agree that the objectives were not well stated and constructed. The paragraph containing the objectives has been completely rewritten (lines 81–97 in revised ms). About the comment between brackets: Yes; that is the idea for future studies: running PCR assays in a sample of the suggested species to find out if any individuals are positive for PaV1. 

10. Lines 68-79: delete this paragraph, superfluous.

R. This paragraph and all references to C. solearis have been deleted

11. Line 86: strike “relatively”. Ecosystem effects are unknown. Line 80: use a hyphen rather than a “/” when discussing host-parasite systems.

R. Changed as suggested.

12. Line 143: why was 50 cm selected as the reference distance between points?

R. Points must be equally spaced along the transect and to have sufficient points per transect so as to be able to estimate percentages of cover. For comparative purposes, we used the same distance between points as Davies et al. (2019) [16].

13. Lines 216-248: which variables are dependent variables and which are independent variables in the multitude of statistical analyses? Spell it out.

R. The main factors are the independent variables. We have now changed the term “main factors” to “independent variables”.

14. Lines 290, 293, 342, and elsewhere: Put emphasis on the finding, not on the statistic. Could rewrite as: Zone A had the lowest average HAS value and Zone B had the highest (ANOVA, Tukey’s HSD). “Results of GLMs showed that” this is an empty phrase. We’re interested in the biological finding.

R. OK. Throughout the Results section, emphasis has now been put in the finding, supported by the statistic. Thanks for the suggestion.

15. Lines 361-368: delete this section on C. solearis. Data are too sparse. Pick up at line 369.

R. The work on C. solearis has been removed from the manuscript.

16. Line 404: the 95% confidence intervals: are these based on the quadrat data or the prevalence? If the latter then it’s a binomial (0/1) and the sd = square root (npq), thus the CIs would be smaller.

R. The 95% confidence intervals are based on prevalence, which is a percentage (i.e., proportion x 100). As explained in lines 246-248 (Materials and Methods): “clinical prevalence (the percentage of clinically infected lobsters) was estimated for each sampling period, and 95% confidence intervals were computed using Wilson’s score method with continuity correction [55]”. = Newcombe (1998).

17. Discussion shows an important problem in the design on line 410: “seemingly different characteristics were selected”. A pilot study or earlier work in the region might have uncovered this issue. The zones weren’t different because the habitat was 73% seagrass in each zone; i.e., they're not very different.

R. Actually, our choice of sampling zones was based on the findings of many earlier studies. This has been clarified in the revised ms (new line 101-115). The Puerto Morelos reef lagoon is a UNESCO’s CARICOMP (Caribbean Coastal Marine Productivity Program) site and has been extensively studied for decades (e.g., van Tussenbroek 1995, 1998, 2011; Ruiz-Rentería et al. 1998; Enríquez and Pantoja-Reyes 2005; Rodríguez-Martínez et al. 2010; van Tussenbroek et al. 2014; Zarco-Perelló and Enríquez 2019) [29–36]. These studies have consistently recognized three lagoon zones based on different characteristics of the vegetation: (1) a narrow coastal fringe, (2) a broad mid-lagoon zone, and (3) an area of back-reef vegetation. In the broad mid-lagoon zone, differences have been found in some areas, such as those where our zones B and C were located (Enríquez and Pantoja Reyes 2005 [33]; van Tussenbroek 2011 [31]). Our zone A was located in the area of back-reef vegetation. Although our zones did not differ in the invertebrate community composition, they did differ in habitat complexity and % cover of most substrates (see Table 1), and some of these characteristics also varied with sampling period. However, unlike in Davies et al. (2019) [16], where differences between a reef lagoon and a back-reef site in Bahía de la Ascensión were larger and related with PaV1 prevalence, the differences in habitat characteristics in our sampling zones at Puerto Morelos did not appear to be sufficiently large so as to influence prevalence of PaV1. 

18. Was temporal variation important? Was spatial variation important? Were there differences in this seemingly homogeneous habitat? These were investigated but they weren't presented well. A bigger question might be were their differences in lobster density within the zones and did this affect prevalence?

R: Both spatial and temporal variation was important for habitat complexity (HAS values) and the abundance (% cover) of substrates such as seagrass, macroalgae, rubble, and sponges, but not for the composition of the invertebrate communities or for PaV1 prevalence. Unfortunately, we cannot provide estimates of lobster density because we do not have an accurate estimate of the area of the zones that were surveilled for lobsters.

19. Lines 436-438: See above points regarding seagrass as an “environmental reservoir”. Also important would be host effects. I can think of many that could contribute to the host surviving longer in areas with more cover, easier access to food, etc. Host factors are important here and the predilection for the smaller animals may not be the only one.

R: On the comment about seagrass as an “environmental reservoir”, please refer to response to comment 7. On the rest of the comment, we are not sure what the reviewer is referring to, as these lines make no reference to host survival.

20. Lines 510: not sure I agree with reference #28, stressors are known to increase pathogens that are host specialists as well as generalists. Vector-borne diseases are good examples. I think the issue is direct vs. indirect life cycles rather than host specificity. Suggest deleting or re-writing.

R. Reference to former reference #28 has been removed from this paragraph.

21. Table 2: the indices really should be labeled in the table heading.

R. OK. Indices have been labeled in the table heading.

22. Figure 2: not sure that this figure contributes much to the Results or Discussion, particularly given Figure 3. Suggest using one or the other, but not both. I’d go with Figure 3 and I’d probably use stacked bar graphs with the means data, but that may not be appropriate.

R. We respectfully disagree. Figs 2 and 3 express different results. It is recommended combining multivariate analyses (considered more as exploratory analyses) and univariate analyses (which are tests of hypotheses). Fig 2 provides information on the relative importance of different types of substrates driving differences among zones over time, whereas Fig 3 shows the mean ± 95% CI for each substrate per zone over time.

23. Figure 4: this might be best as supplemental data. There is very little variation in this data and I wonder if the statistical analysis had enough power to give any credence to the significant differences shown. This should be discussed.

R. We have moved Fig 4 to supporting information. However, we are not sure about the reference to power. Fig 4 (now S1 Fig) shows the mean ± 95% CI of HAS, by zone and period. In the GLM, the effect of both factors (i.e., independent variables) was significant; therefore, the null hypothesis (of no difference among means) was rejected. Power analysis is recommended when the null hypothesis is not rejected, but even in those cases some statisticians (e.g., Hoenig & Heisey 2001) argue against its use and in favor of putting more emphasis on the investigator's choice of hypotheses and on the interpretation of confidence intervals.

24. Figure 5: I would use either Figure 4 or Figure 5 (I prefer Fig 5) to show that there is little separation in the habitats. Could do this for infected vs. uninfected animals and see if there are differences?

R. Fig 4 (mean ± 95% CI of HAS, by zone and period) has been moved to supporting information (new S1 Fig). However, Fig 5 depicts the results of multivariate analyses on invertebrate community composition, so we are not sure what the reviewer means by “could do this for infected vs uninfected animals and see if there are differences?” Unless the reviewer means something similar to former Fig 4, in which case please see response to comment 26 below.

25. Figure 6: the clinical prevalence is lower than might be expected here, only hitting 4% at 30 mm CL. By the way, is a 73 mm CL lobster still a juvenile?

R. The overall average clinical prevalence was 16.2%. In Fig 6 (now Fig 5), the entire orange section represents all clinically infected lobsters. The figure shows a mode in the 30 mm CL size class, with 18.3% of the total lobster sample, of which about 1/5 were diseased (i.e., 3.8% of the entire lobster sample). Along the Mexican Caribbean P. argus lobsters mature at about 75 to 80 mm CL (see Fonseca-Larios and Briones-Fourzán 1998), so a 73 mm CL lobster is still sexually immature (i.e., juvenile). 

26. I would add an additional figure here. Shouldn’t there be a figure showing prevalence of PaV1 by site*time. (I was expecting something akin to Figure 7 but with prevalence data). I know this has been done in other studies of PaV1 from the region.

R. The first logistic model showed that probability of clinical infection was not affected by zone; i.e., the significant differences in habitat complexity and in % cover of substrates among zones were not related with the prevalence of PaV1. Therefore, it was possible to pool data of lobsters from all zones to have a global, local estimation of prevalence. Regardless, an additional figure of prevalence akin to former Fig 7 is now in the supporting information (S4 Fig), where former Fig 4 (now S1 Fig) and Fig 7 (now S3 Fig) were moved, as suggested by the reviewer.

27. Figure 7 could be deleted with no loss to the main points.

R. OK. Fig 7 has been moved to supporting information (now S3 Fig). 

28. Figure 8: see comments regarding the estimation of standard deviations from binomial data. I presume these are mean values from transects or zones, grouped and analyzed collectively. If that’s the case they are not binomial data, and the sample size isn’t reflective of the zone or transect number.

R. The 95% confidence intervals for prevalence were not estimated from binomial data (i.e. 1/0); they were estimated from binomial proportional data (percentages). See Newcombe (1998) [55] or Agresti & Coull (1998). As mentioned in the previous response, results of the logistic regression models made it possible to pool data of lobsters from all zones to have a global, local estimation of prevalence. Pooling the lobsters from all zones has the advantage of increasing sample size, which decreases the width of the 95% CI, thus providing a more accurate estimation of prevalence for monitoring purposes over the longer term.

Minor style points:

29. Line 28: “and/or”. Just use one or the other, not both. “Or” usually works best. 

R: Sentence had been rewritten.

30. Line 51: “structured crevice-type shelters”. This is jargon. It’s structured habitats or crevice-like shelters or dens. 

R: OK, “structured” has been removed.

31. Line 149: Re the habitat complexity estimates: these are probably more subjective than qualitative. 

R: The estimates were made by scientists who have been doing them for several years, so though qualitative, they are not subjective.

32. Line 162: give minimum size of animals identified in the biodiversity component. Was it >10 cm or >10 mm or some other value? 

R: The minimum size has been added (>1 cm).

33. Line 199: American spelling was used throughout except for “Haemolymph”. Consistency. 

R: OK. Thanks for catching this.

34. Line 216: “percent data on the cover” is jargon. “data on the percent cover” is not. For statistical purposes, in logit-transformed data, I presume it’s presence/absence (0/1) data that are being analysed as in a logistic regression?

R: Sentence has been changed to “data on percent cover”. In this case, the data being analyzed is the percentage of cover (not the presence/absence) of the different substrates. Percentages (or proportions) are not normally distributed and need to be properly transformed to be analyzed. The arcsine square root transformation is the most common transformation of proportions, but Warton & Hui (2011) [48] argue that it is much better to transform proportions to logits. The logit transform of a proportion is: log (p/[1–p]) (where p = proportion). However, because the logit transform for proportions equal to 0 and 1 are the undefined values -� and �, respectively, an ad hoc solution is to add some small value � to both the numerator and denominator of the logit function. Warton & Hui [48] propose taking as � the minimum non-zero proportion (or if proportions are large, the minimum non-zero value for 1–p). 

35. Line 233: why was square-root transformation used? I know why, but mention briefly to readers, i.e., to adjust variance to meet assumptions of normality.

R: Transformation serves a different purpose in multivariate analyses for communities than in univariate analyses. In biological communities, some species may be very abundant and some may be rare. This affects the computation of the similarity (Bray-Curtis) index matrix, but may be alleviated by transforming the data. In ordination analyses (e.g. nMDS), the data transform sequence is: no transform, square root, fourth root (or log (x+1)), and presence/absence. These transformations shift the focus from the dominant species (no transform) to the rarer species (presence/absence), so when one has quantitative data (and not only presence/absence data), the root or fourth root transforms are recommended. These transforms have the effect of down-weighting the importance of the highly abundant species, so that similarities depend not only on their values but also on less common species. In other words, the root transform retains the quantitative information while downplaying the species dominants (Clarke & Warwick 2001) [53]. This last sentence has been added in the revised ms (new lines 233-234).

36. Line 494-495: rewrite this sentence. There are well known means to adjust clinical prevalence levels using sensitivity and specificity.

R. Yes there are, but as shown by Pestal et al. (2003), in the special case where specificity equals 1.0, then the estimation of the proportion of “truly infected” organisms is reduced to the estimated prevalence based on macroscopic criteria over the sensitivity estimate. In this case, the sensitivity of clinical signs was 0.5 (i.e., for each lobster with clinical signs there is another one infected but with no clinical signs) and specificity was 1.0 (i.e., all lobsters with clinical signs were positive for PaV1 by PCR. Therefore, multiplying clinical prevalence by 2 can provide a rough estimation of real prevalence (Huchin-Mian et al. 2013 [15]; Candia-Zulbarán et al. 2019 [46]).

37. Line 546: should spell out this journal reference.

R. This and other references on Cymatocarpus solearis have been removed.

Reviewer #2: Summary:

This manuscript describes an ambitious study aimed at determining whether there are habitat or community characteristics that can explain the prevalence of the virus PaV1 or the digenean trematode Cymatocarpus solearis among Caribbean spiny lobsters in a tropical reef lagoon offshore of Puerto Morelos, Mexico. The authors did not find any consistent associations between any of the characteristics they measured and the prevalence of these pathogens, other than one spike in PaV1 prevalence during one of the sampling periods.

General comments:

Overall the manuscript was well written and well organized. The abstract needs revision and the introduction needs some reorganization and revision (marked copy of these sections is attached), but the remainder of the manuscript text was clear and easily interpreted. The approach seemed appropriate, as were the statistical analyses and interpretation. The main problem with the manuscript is the lack of any consequential, significant findings. The only significant finding was a bump up in prevalence of PaV1 during one sampling period and this does not create any kind of pattern for interpretation (and the authors recognized this). This is always a difficult position to be in and I applaud the authors for writing the paper regardless because the lack of patterns or associations is still important to publish because, if for nothing else, it keeps other researchers for attempting similar studies that are apt to find the same results.

1. The sampling zones seem rather close to one another. I wonder if the proximity of the zones to one another doesn’t allow the lobsters to easily move between them and effectively ameliorate any effect of habitat or community characteristics on prevalence of these pathogens? I suggest that this be addressed in the discussion.

R. Thanks for calling our attention to this potential misinterpretation of the data. The sampling zones were chosen based on different attributes of the reef lagoon benthic habitats (see response 17 to Reviewer 1). The population of P. argus in the reef lagoon comprises mainly juveniles and the distance between zones ranged between 600 m and 1 km. These distances exceed the movement ranges of juvenile P. argus ≤50 mm CL as assessed in sites enhanced with casitas in this same reef lagoon (<100 m) (Briones-Fourzán et al. 2007) [27]. This information has been added in the revised manuscript.

Specific comments:

2. PDF is attached with suggested edits to title, abstract, and introduction. Remainder of the manuscript was much better.

R. Suggestions in the PDF attached by Reviewer 2 have been taken into account. The title has been changed, and the abstract and introduction have undergone substantial rewriting. All mention to the work on C. solearis was removed from the manuscript.

3. Note: ignore the bracketed comment on lines 68-79. At that point I thought the research was focused solely on PaV1 since C. solearis is not mentioned in the abstract at all (fix that)!

R. The work on C. solearis was removed from the manuscript to focus on the PaV1 results.

4. Line 172 Give a rationale for including the additional lobster sampling periods

R. OK. Lobster samplings were continued in order to monitor the prevalence of the disease in this reef lagoon system over the long term. This has now been clarified in the revised manuscript.

5. It’s confusing the way the tables are nested in the manuscript with the captions following them, and the figure captions within the manuscript text. Are they not supposed to be after the literature cited?

R. Nesting the tables in the manuscript with the captions following them, and putting the figure captions where the figures are suggested to be placed, are requested by PLoS ONE in the Instructions to Authors.

6. Figures and table are nice and clearly constructed.

R. Thank you!

WE THANK REVIEWERS 1 AND 2, AND THE ACADEMIC EDITOR FOR THEIR THOROUGH REVIEWS AND VALUABLE COMMENTS

6. PLOS authors have the option to publish the peer review history of their article (what does this mean?). If published, this will include your full peer review and any attached files.

Do you want your identity to be public for this peer review? For information about this choice, including consent withdrawal, please see our Privacy Policy.

Reviewer #1: No

Reviewer #2: No

References to papers cited in our Responses that are not in the revised manuscript:

Agresti A, Coull BA. 1998. Approximate is better than "Exact" for interval estimation of binomial proportions. Am Stat 52: 119–126.

Fonseca-Larios, Briones-Fourzán P. 1998. Fecundity of the spiny lobster Panulirus argus (Latreille, 1804) in the Caribbean coast of Mexico. Bull Mar Sci 63: 21–32. 

Hoenig JM, Heisey DM. 2001. The abuse of power: the pervasive fallacy of power calculations for data analysis. Am Stat 55: 1–6.

Pestal GP, Taylor DM, Hoenig JM, Shields JD, Pickavance R. 2003. Monitoring the prevalence of the parasitic dinoflagellate Hematodinium sp. in snow crabs Chionoecetes opilio from Conception Bay, Newfoundland. Dis Aquat Org 53: 67−75.

Sweet MJ, Bythell JC, Nugues MM. 2013. Algae as reservoirs for coral pathogens. PLoS ONE 8(7): e69717.

Raymond B, Hartley SE, Cory JS, Hails RS. 2005. The role of food plant and pathogen-induced behaviour in the persistence of a nucleopolyhedrovirus. J Invertebr Pathol 88: 49–57.

---

## [Decision Letter · Decision Letter 1]

3 Feb 2020

PONE-D-19-25184R1

Ecological determinants in the prevalence of Panulirus argus virus 1 (PaV1) in juvenile Caribbean spiny lobsters in a tropical reef lagoon

PLOS ONE

Dear Dr. Briones-Fourzán,

Thank you for submitting your manuscript to PLOS ONE. After careful consideration, we feel that it has merit but does not fully meet PLOS ONE’s publication criteria as it currently stands. Therefore, we invite you to submit a revised version of the manuscript that addresses the points raised during the review process.

I think the authors have done a good job addressing most of the comments of the reviewers and the reviewer generally agreed (note that the second reviewer was not available to assess the revision). The reviewer and I have both provided some minor comments. However, I agree with the reviewer in that it is possible, if not likely, that the design of the study contributed to the results and a future study should use sites that are much further apart. Both reviewers made this comment during the first round of review, and I agree, so the authors need to address it in some way in the manuscript.  I think at the very least the authors need to acknowledge this as a potential issue with the current study and advice for a future study. 

We would appreciate receiving your revised manuscript by Mar 19 2020 11:59PM. To enhance the reproducibility of your results, we recommend that if applicable you deposit your laboratory protocols in protocols.io, where a protocol can be assigned its own identifier (DOI) such that it can be cited independently in the future. For instructions see: http://journals.plos.org/plosone/s/submission-guidelines#loc-laboratory-protocols

We look forward to receiving your revised manuscript.

Kind regards,

Heather M. Patterson, Ph.D.

Academic Editor

PLOS ONE

Reviewers' comments:

Reviewer's Responses to Questions

**Comments to the Author**

1. If the authors have adequately addressed your comments raised in a previous round of review and you feel that this manuscript is now acceptable for publication, you may indicate that here to bypass the “Comments to the Author” section, enter your conflict of interest statement in the “Confidential to Editor” section, and submit your "Accept" recommendation.

Reviewer #2: (No Response)

2. Is the manuscript technically sound, and do the data support the conclusions?

Reviewer #2: Partly

3. Has the statistical analysis been performed appropriately and rigorously? 

Reviewer #2: Yes

4. Have the authors made all data underlying the findings in their manuscript fully available?

Reviewer #2: Yes

5. Is the manuscript presented in an intelligible fashion and written in standard English?

Reviewer #2: Yes

6. Review Comments to the Author

Reviewer #2: The authors addressed most of my concerns noted in the first review; however, the core problems still exist. The zones that the authors used in the study were rather homogeneous to start and too close together so the lack of a difference in PaV1 was really not surprising. I think the authors did a decent job recognizing the former in the discussion, so I am OK with that. Their other responses to my comments and suggestions are also acceptable.

However, in their response to my inquiry about lobster movement between zones, the authors reference a study that shows small juvenile lobsters have a foraging range of < 50m as justification for the proximity of the zones making them independent. As I understand it from that paper, that foraging range is an estimate for a single night (straight line distance from the shelter used one night to the one used on the next night). Thus, over the course of weeks a juvenile lobster could easily shelters as steppingstones to move between the habitat zones in the study, and therefore, move PaV1 between zones. The authors need to mention more explicitly that the proximity between the zones (A & B appear < 500m apart) may have allowed lobsters to mingle and ameliorate the effect of any difference between zones, or something to that effect.

Specific comments:

The title seems to suggest that ecological determinants were found when they really were not. I suggest the title be changed to a question. “Do ecological characteristics drive the prevalence of…?” I say this because I think the problems with the design (proximity between zones and their ecological similarity) might really be why the ecological characteristics that were assessed where not significant drivers of PaV1, not that they might not be significant in another system/design.

The detailed info on the monitoring of PaV1 in the lagoon should be omitted from the abstract. Keep that to one general line.

7. PLOS authors have the option to publish the peer review history of their article (what does this mean?). If published, this will include your full peer review and any attached files.

Reviewer #2: No

---

## [Author Response · Author response to Decision Letter 1]

13 Feb 2020

PONE-D-19-25184R1

Ecological determinants in the prevalence of Panulirus argus virus 1 (PaV1) in juvenile Caribbean spiny lobsters in a tropical reef lagoon

PLOS ONE

Dear Dr. Briones-Fourzán,

Thank you for submitting your manuscript to PLOS ONE. After careful consideration, we feel that it has merit but does not fully meet PLOS ONE’s publication criteria as it currently stands. Therefore, we invite you to submit a revised version of the manuscript that addresses the points raised during the review process.

I think the authors have done a good job addressing most of the comments of the reviewers and the reviewer generally agreed (note that the second reviewer was not available to assess the revision). The reviewer and I have both provided some minor comments. However, I agree with the reviewer in that it is possible, if not likely, that the design of the study contributed to the results and a future study should use sites that are much further apart. Both reviewers made this comment during the first round of review, and I agree, so the authors need to address it in some way in the manuscript. I think at the very least the authors need to acknowledge this as a potential issue with the current study and advice for a future study.

R: The potential issue has been acknowledged. Please see response to comments by Reviewer # 2 below.

We would appreciate receiving your revised manuscript by Mar 19 2020 11:59PM. To enhance the reproducibility of your results, we recommend that if applicable you deposit your laboratory protocols in protocols.io, where a protocol can be assigned its own identifier (DOI) such that it can be cited independently in the future.

For instructions see: http://journals.plos.org/plosone/s/submission-guidelines#loc-laboratory-protocols

A rebuttal letter that responds to each point raised by the academic editor and reviewer(s). This letter should be uploaded as separate file and labeled 'Response to Reviewers'.

A marked-up copy of your manuscript that highlights changes made to the original version. This file should be uploaded as separate file and labeled 'Revised Manuscript with Track Changes'.

An unmarked version of your revised paper without tracked changes. This file should be uploaded as separate file and labeled 'Manuscript'.

We look forward to receiving your revised manuscript. Kind regards,

Heather M. Patterson, Ph.D. Academic Editor

PLOS ONE

Reviewers' comments:

Reviewer's Responses to Questions

Comments to the Author

1. If the authors have adequately addressed your comments raised in a previous round of review and you feel that this manuscript is now acceptable for publication, you may indicate that here to bypass the “Comments to the Author” enter your conflict of interest statement in the “Confidential to Editor” section, and submit your "Accept" recommendation. Reviewer #2: (No Response)

2. Is the manuscript technically sound, and do the data support the conclusions?

Reviewer #2: Partly

3. Has the statistical analysis been performed appropriately and rigorously? Reviewer #2: Yes

4. Have the authors made all data underlying the findings in their manuscript fully available?

Reviewer #2: Yes

5. Is the manuscript presented in an intelligible fashion and written in standard English?

Reviewer #2: Yes

6. Review Comments to the Author

Please use the space provided to explain your answers to the questions above. You may also include additional comments for the author, including concerns about dual publication, research ethics, or publication ethics. (Please upload your review as an

attachment if it exceeds 20,000 characters)

Reviewer #2: The authors addressed most of my concerns noted in the first review; however, the core problems still exist. The zones that the authors used in the study were rather homogeneous to start and too close together so the lack of a difference in PaV1 was really not surprising. I think the authors did a decent job recognizing the former in the discussion, so I am OK with that. Their other responses to my comments and suggestions are also acceptable.

Response: Thank you for the positive feedback.

However, in their response to my inquiry about lobster movement between zones, the authors reference a study that shows small juvenile lobsters have a foraging range of < 50m as justification for the proximity of the zones making them independent. As I understand it from that paper, that foraging range is an estimate for a single night (straight line distance from the shelter used one night to the one used on the next night). Thus, over the course of weeks a juvenile lobster could easily shelters as steppingstones to move between the habitat zones in the study, and therefore, move PaV1 between zones. The authors need to mention more explicitly that the proximity between the zones (A & B appear < 500m apart) may have allowed lobsters to mingle and ameliorate the effect of any difference between zones, or something to that effect.

Response: [Just to clarify: In the referenced study (Briones-Fourzán et al. 2007), the foraging range was not an estimate for a single night. It was an estimate based on subsequent recaptures of marked lobsters, which occurred during surveys conducted at two to three-month intervals. Due to the scarcity of natural shelters in the reef lagoon, the average persistence of juveniles <50 mm CL was twice as long in sites with casitas (up to 3 months) as in sites with no casitas (up to 1.5 months).]

In the Discussion of the previous ms, after comparing our results with those obtained in Bahía de la Ascensión, we wrote “Therefore, it would appear that the scale of habitat differences required to be associated with a change in disease prevalence requires a larger range of lobster sizes or a wider variation in habitats, such as those studied in Bahía de la Ascensión.” But distance was not explicitly mentioned. Therefore, we have now added the following paragraph (lines 414-421 of revised ms): “Because natural crevice-type shelters for lobsters are very scarce in the Puerto Morelos reef lagoon [39], the sampling zones included experimental sites where casitas were deployed years ago for other studies [25,27,39,40]. Casitas increase density of juvenile lobsters as well as their persistence in a site [27], and the distance between our sampling zones was greater than the average movement ranges of juvenile P. argus [2,26,27]. Yet, it cannot be dismissed that some mingling of lobsters could occur over time, potentially masking any effect of habitat characteristics on PaV1 prevalence. Therefore, future studies should use sampling sites that are further apart and, whenever possible, located over more heterogeneous habitats.” This involved shortening the third paragraph (between lines 449 and 451) after the new one to avoid repetition.

Specific comments:

The title seems to suggest that ecological determinants were found when they really were not. I suggest the title be changed to a question. “Do ecological characteristics drive the prevalence of…?” I say this because I think the problems with the design (proximity between zones and their ecological similarity) might really be why the ecological characteristics that were assessed where not significant drivers of PaV1, not that they might not be significant in another system/design.

Response: Title has been changed as suggested. We also changed the running title. 

The detailed info on the monitoring of PaV1 in the lagoon should be omitted from the abstract. Keep that to one general line.

Response: The abstract has been modified as suggested. 

7. PLOS authors have the option to publish the peer review history of their article (what does this mean?). If published, this will include your full peer review and any attached files.

Do you want your identity to be public for this peer review? For information about this choice, including consent withdrawal, please see our Privacy Policy.

Reviewer #2: No

---

## [Editor Report · Decision Letter 2]

18 Feb 2020

Do ecological characteristics drive the prevalence of Panulirus argus virus 1 (PaV1) in juvenile Caribbean spiny lobsters in a tropical reef lagoon?

PONE-D-19-25184R2

Dear Dr. Briones-Fourzán,

We are pleased to inform you that your manuscript has been judged scientifically suitable for publication and will be formally accepted for publication once it complies with all outstanding technical requirements.

With kind regards,

Heather M. Patterson, Ph.D.

Academic Editor

PLOS ONE
---

## [Editor Report · Acceptance letter]

21 Feb 2020

PONE-D-19-25184R2 

Do ecological characteristics drive the prevalence of *Panulirus argus* virus 1 (PaV1) in juvenile Caribbean spiny lobsters in a tropical reef lagoon? 

Dear Dr. Briones-Fourzán:

I am pleased to inform you that your manuscript has been deemed suitable for publication in PLOS ONE. Congratulations! Your manuscript is now with our production department. 

With kind regards,

on behalf of

Dr. Heather M. Patterson 

Academic Editor

PLOS ONE